# GRT-R910: a self-amplifying mRNA SARS-CoV-2 vaccine boosts immunity for ≥6 months in previously-vaccinated older adults

Christine D. Palmer [1], Ciaran D. Scallan[1], Lauren D. Kraemer Tardif[1], Melissa A. Kachura [1], Amy R. Rappaport[1], Daniel O. Koralek[1], Alison Uriel[2], Leonid Gitlin[1], Joshua Klein[1], Matthew J. Davis[1], Harshni Venkatraman[1], Meghan G. Hart[1], Jason R. Jaroslavsky [1], Sonia Kounlavouth[1], Martina Marrali [1], Charmaine N. Nganje[1], Kyounghwa Bae[1], Tiffany Yan[1], Katharyn Leodones[1], Milana Egorova [1], Sue-Jean Hong[1], Jenchun Kuan[1], Silvia Grappi[3], Pedro Garbes [1], Karin Jooss [1] ✉ & Andrew Ustianowski[2]

SARS-CoV-2 has resulted in high levels of morbidity and mortality world-wide, and severe complications can occur in older populations. Humoral immunity induced by authorized vaccines wanes within 6 months, and frequent boosts may only offer transient protection. GRT-R910 is an investigational self-amplifying mRNA (samRNA)-based SARS-CoV-2 vaccine delivering full-length Spike and selected conserved non-Spike T cell epitopes. This study reports interim analyses for a phase I open-label dose-escalation trial evaluating GRT-R910 in previously vaccinated healthy older adults (NCT05148962). Primary endpoints of safety and tolerability were assessed. Most solicited local and systemic adverse events (AEs) following GRT-R910 dosing were mild to moderate and transient, and no treatment-related serious AEs were observed. The secondary endpoint of immunogenicity was assessed via IgG binding assays, neutralization assays, interferon-gamma ELISpot, and intracellular cytokine staining. Neutralizing antibody titers against ancestral Spike and variants of concern were boosted or induced by GRT-R910 and, contrasting to authorized vaccines, persisted through at least 6 months after the booster dose. GRT-R910 increased and/or broadened functional Spike-specific T cell responses and primed functional T cell responses to conserved non-Spike epitopes. This study is limited due to small sample size, and additional data from ongoing studies will be required to corroborate these interim findings.

The COVID-19 pandemic caused by the Severe Acute Respiratory Syndrome Coronavirus 2 (SARS-CoV-2) has resulted in high levels of morbidity and mortality throughout the world, accounting for more than 600 million cases and 6.5 million deaths by October 2022[1]. While many of those infected only develop either mild respiratory symptoms or remain asymptomatic, severe complications such as pneumonia, acute respiratory distress syndrome, and death can occur, particularly in populations ≥60 years of age and those with certain co-morbidities[2]. Although the rollout of authorized vaccines has helped reduce the severity, morbidity, and mortality associated with COVID-19[3,4],

[1]Gritstone bio, Inc., Emeryville, CA, USA. [2]North Manchester General Hospital & University of Manchester, Manchester, UK. [3]VisMederi Srl., Siena, Italy. ✉e-mail: kjooss@gritstone.com

humoral immunity induced by currently authorized vaccines wanes within 6 months[5], leading to increased rates of breakthrough infections[6]. Whereas administration of a third dose proved effective in boosting immunity[7], frequent boosts offer only transient protection against re-infection[8,9] and may not be a successful long-term strategy for public health. Newly emerging SARS-CoV-2 variants continue to pose a population risk due to increased transmissibility[10], and after the emergence of the Omicron variant, vaccine effectiveness of three doses of BNT162b2 or mRNA-1273 waned within four to five months as neutralizing antibody titers declined[11]. New SARS-CoV-2 sub-variants continue to be identified that may partially or fully evade immunity induced by authorized vaccines[12,13] and monoclonal antibodies[14], leading to increased COVID-19 incidence rates (European Centre for Disease Prevention and Control). Thus, there is still an unmet need for additional SARS-CoV-2 vaccines that can provide long-term durability and protection against severe disease across different variants, especially in more vulnerable populations such as those ≥60 years of age. Currently authorized mRNA-based COVID-19 vaccines exclusively target the Spike (S) protein, which is highly susceptible to mutations, increasing the risk of immune evasion as new variants emerge. Vaccine-induced T cell responses to S protein contributed to protection in non-human primates (NHPs)[15], and pre-existing cross-reactive T cell responses to non-structural coronavirus components were associated with clearance of subclinical SARS-CoV-2 infection in frequently exposed healthcare workers and exposed household contacts[16,17]. A new vaccine encoding for T cell epitopes, including sequences from SARS-CoV-2 proteins outside of Spike, could enhance the immune response against future variants of concern (VOC) and offer more long-lasting protection[18].

GRT-R910 is an investigational vaccine that was designed to enhance the immune response, particularly against potential future SARS-CoV-2 VOC. GRT-R910 utilizes a fully synthetic Venezuelan Equine Encephalitis Virus (VEEV)-based self-amplifying mRNA (samRNA) vector formulated into lipid nanoparticles (LNP), a platform that has been safe and immunogenic in cancer patients[19] and drove protective immunity against SARS-CoV-2 infection in NHPs[20]. The amplifying nature of samRNA vaccines generates high and durable transgene expression compared to mRNA vaccines[21], thereby driving high nAb titers in mice[22,23] for a long period of time at low doses, suggesting that this platform can be dose-sparing[20]. GRT-R910 encodes for full-length S prefusion modified from the original Wuhan Hu-1 strain with a D614G mutation ($S_{D614G}$) and highly conserved non-S T cell epitope (TCE) sequences predicted using a previously published algorithm that allows for accurate prediction of HLA-peptide complexes[24] combined with selected validated epitopes[25,26]. These epitopes were selected to provide broad HLA-coverage across major human populations and include conserved regions across coronaviruses. In this study, we report an interim analysis for GO-009 (NCT05148962), a phase I open label dose-escalation trial conducted in the UK designed to evaluate the safety, tolerability, and immunogenicity of GRT-R910 in previously vaccinated healthy adults ≥60 years of age, with expansion of cohorts to adults ≥18 years of age ongoing. Here, we describe interim results for the primary endpoints of safety and tolerability and the secondary endpoint of immunogenicity over a period of 6 months following GRT-R910 booster in the initial two dose-escalation cohorts with participants ≥60 year of age who previously received AZD1222 as a primary 2-dose series.

## Results

### Vaccine and trial design, demographics, and enrollment
GO-009 (NCT05148962) is a multi-center, open-label, ongoing phase I study to assess the safety (primary endpoint) and immunogenicity (secondary endpoint) of GRT-R910 (Fig. 1A), a SARS-CoV-2 samRNA vaccine expressing full-length prefusion modified $S_{D614G}$ and selected TCE frames from conserved SARS-CoV-2 genes, in healthy adults ≥60

years of age who had previously completed a primary vaccination series with AstraZeneca ChAdOx1 AZD1222. A single 10 μg or 30 μg dose of GRT-R910 samRNA, followed by an optional second dose approximately 4 months later at the same dose-level, was initially assessed in cohorts 1 and 2 (Fig. 1B). At time of data cut-off, 54 participants had been assessed for eligibility, with 33 participants receiving vaccination in cohorts 1-4 (Fig. 1C). This interim report describes 6-month data for cohorts 1 and 2 only; data accrual for cohorts 3-6 is ongoing. For cohort 1 (10 μg), 10 participants received dose 1 of GRT-R910, and 6 of those subsequently elected to receive a second 10 μg dose approximately 20 weeks later. Seven participants received the first dose of GRT-R910 samRNA in cohort 2 (30 μg), and four of those participants elected to receive an optional second 30 μg dose. There are no significant differences between cohorts 1 and 2 in terms of age, body mass index (BMI), race, and ethnicities. Cohort 1 includes more male participants (60%) compared to cohort 2 (29%; Table 1). Of note, although 8/17 participants dosed with GRT-R910 ($S_{D614G}$) were diagnosed with SARS-CoV-2 infection within 6 months of dosing (median 3.6 months; range 12 days to 5.2 months) during the emergence of Omicron in 2022[12], none of these participants, who were at higher risk of severe infection (≥60 years of age), required hospitalization, and SARS-CoV-2 symptoms were generally mild.

### Administration of GRT-R910 is well-tolerated in previously vaccinated healthy adults ≥ 60 years of age who previously received two doses of AZD1222
The primary endpoint of safety and tolerability were assessed. All vaccine regimens were well-tolerated, and most of the solicited local and systemic AEs within 7 days of each dose (10 and 30 μg) of GRT-R910 were mild or moderate and transient, and no treatment-related serious AEs (SAEs) were observed (Fig. 1D). Solicited local AEs occurring in >10% of the participants included erythema, edema, tenderness, and pain. Solicited systemic AEs occurring in >10% of the participants included nausea, headache, arthralgia, myalgia, fatigue, fever, and chills. Solicited local and systemic AEs typically resolved within 24–48 h. No grade 3 events were observed in cohort 1 (10 μg). Solicited AEs ≥ 8 days and unsolicited AEs ≤ 28 days are summarized in Table 2.

### GRT-R910 induces cross-reactive binding antibody levels to Spike VOC that are maintained for 6 months
Humoral immunity against early SARS-CoV-2 S variants Wuhan-Hu-1 ($S_{WT}$), Victoria/01/2020 ($S_{VIC}$), $S_{D614G}$ and subsequent VOC $S_{Beta}$ (B.1.351), $S_{Delta}$ (B.1.617.2), and $S_{Omicron}$ (BA1 and BA5) was assessed as part of the secondary outcome measures for this interim analysis. Sera were analyzed for $S_{WT}$-specific IgG levels pre and post-administration of 10 or 30 μg of GRT-R910 samRNA via ELISA. No clear dose-response was observed when comparing S-specific IgG levels between dose cohorts at post-vaccination timepoints (D29 and D57; Supplementary Fig. S1A). As expected, Nucleocapsid (N)-specific IgG responses were not induced by GRT-R910 samRNA administration and were only detected above pre-pandemic reference range in samples following positive SARS-CoV-2 diagnosis while on study (Supplementary Fig. S2). Administration of a single dose of GRT-R910 boosted the $S_{WT}$-specific IgG levels 12.8- and 13.4-fold 4- and 8-weeks post administration from 708.6 ELU/ml (GeoMean) at baseline (D1; 19–31 weeks post AZD1222 primary vaccination series) to 9085 ELU/ml and 9514 ELU/ml at D29 and D57, respectively (n = 17, Fig. 2A). In participants who did not receive an optional second dose of GRT-R910 (n = 7), $S_{WT}$-specific IgG levels were maintained at 9005 ELU/ml (GeoMean) through D180 (6 months post GRT-R910 dose; Fig. 2B). Similar longevity of the $S_{WT}$-specific IgG response was observed in participants receiving an optional second dose of GRT-R910, with IgG levels remaining at 6427 ELU/ml (GeoMean) through D293 (6 months post the second dose of GRT-R910; Fig. 2C). Additional assays to assess levels of IgG binding to $S_{D614G}$ and VOCs (Beta, Delta, Omicron

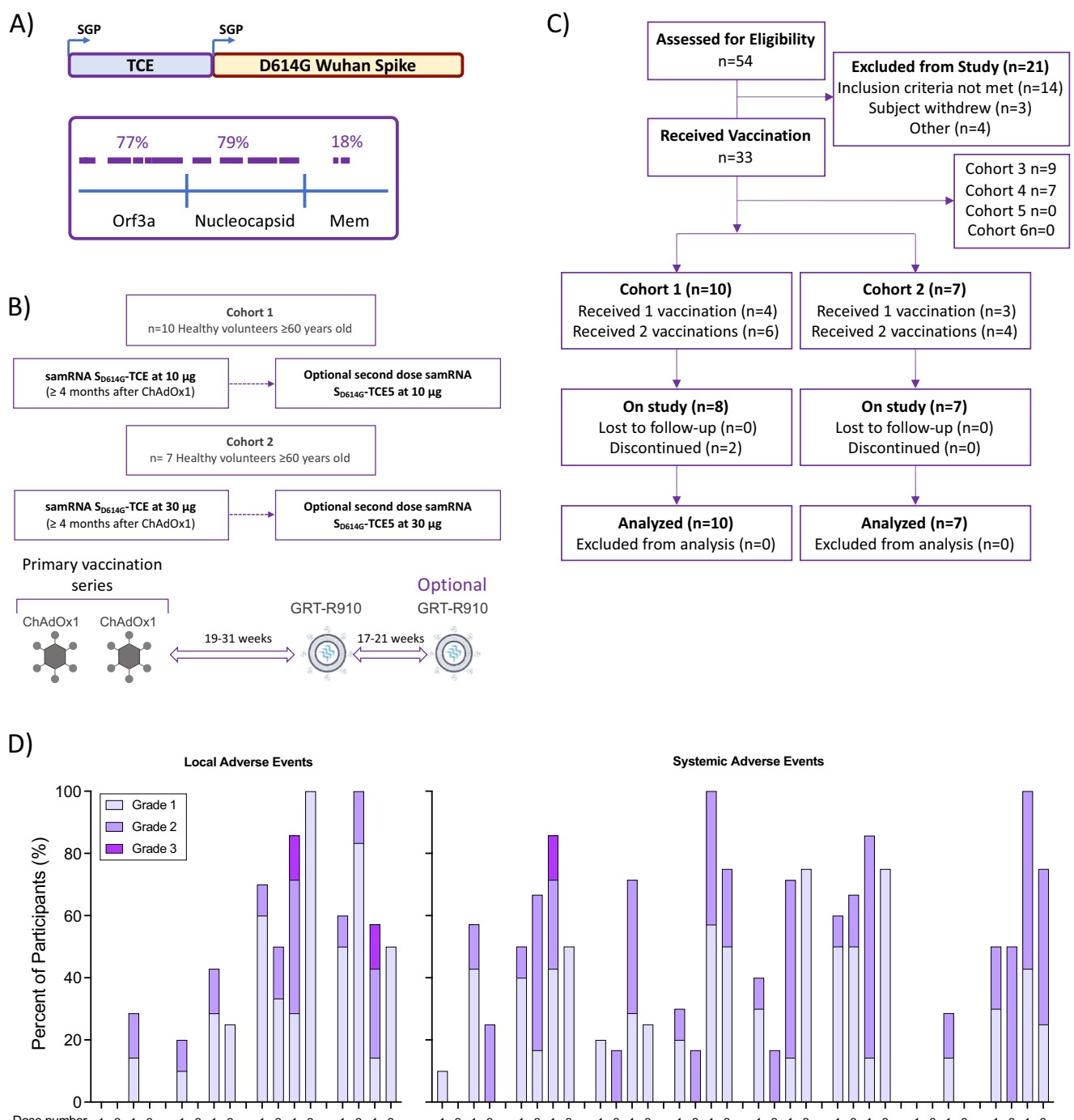

**Fig. 1 | Administration of 1 or 2 doses of samRNA at 10 or 30 μg is well-tolerated by previously vaccinated healthy adults ≥ 60 years of age. A** Schematic representation of GRT-R910 encoding full length Spike$_{D614G}$ and T cell Epitopes (TCE). TCE frames represent 77% of SARS-CoV-2 ORF3a, 79% of Nucleocapsid, and 18% of Membrane sections. **B** Schematic outlining study design for dose-escalation cohorts. **C** CONSORT diagram outlining screening, enrollment, and dosing as of data cut-off August 23, 2022. **D** Local and systemic reactogenicity within 7 days from the first and optional second dose for all participants dosed (Cohort 1 (10 μg dose), $n = 10$; Cohort 2 (30 μg dose), $n = 7$).

BA1) were performed via MSD multiplex assay. To assess equivalence of the ELISA versus the MSD assay for IgG analyses, the correlation between S$_{WT}$ IgG levels in ELU/ml and in arbitrary units/ml measured by ELISA and MSD, respectively, was assessed. Values correlated significantly between the two assays (Supplementary Fig. S1B; Pearson correlation, $n = 88$, $p < 0.001$, $r = 0.88539$), confirming consistent measurement of S$_{WT}$-specific IgG levels with two different assays performed in two different laboratories. Cross-reactive binding to

VOC and longevity of S-specific IgG responses for 6 months following a single dose of GRT-R910 was assessed for S$_{WT}$ and VOCs Beta, Delta, and Omicron BA1. Figure 2D shows an initial boost of IgG levels after a single dose of GRT-R910 administration, followed by long-term maintenance of IgG levels for 6 months (Supplementary Dataset 1). Longevity of S$_{WT}$-specific IgG responses was also assessed in participants who received an optional second dose of GRT-R910 approximately 20 weeks (D113) post initial GRT-R910 dose at D1

**Table 1 | Demographics**

| Characteristics | Cohort 1 GRT-R910 10 µg (N = 10) | Cohort 2 GRT-R910 30 µg (N = 7) |
|---|---|---|
| **Sex, number of participants (%)** | | |
| Male | 6 (60.0) | 2 (28.6) |
| Mean age (range) - year | 68.8 (63–81) | 66.7 (61–74) |
| **BMI (kg/m²)[a]** | | |
| Mean (range) | 26.4 (21.0–34.7) | 27.6 (21.8–35.2) |
| **Study Vaccine Administration, number of participants (%)** | | |
| 1 dose | 4 (40.0) | 3 (42.9) |
| 2 doses | 6 (60.0) | 4 (57.1) |
| **Race & Ethnicity, no. of participants (%)** | | |
| White | 8 (80.0) | 6 (85.7) |
| Asian | 2 (20.0) | 0 |
| Other | 0 | 1 (14.3) |
| Not Hispanic or Latino | 10 (100.0) | 7 (100.0) |

[a]BMI = (body weight in kilograms)/(height in meters)².

**Table 2 | Summary of Solicited (≥ 8 days) and Unsolicited AEs (≤28 days)**

| | Cohort 1 GRT-R910 10 µg | | Cohort 2 GRT-R910 30 µg | |
|---|---|---|---|---|
| | 1st Dose (N = 10) n (%) | 2nd Dose (N = 6) n (%) | 1st Dose (N = 7) n (%) | 2nd Dose (N = 4) n (%) |
| **Solicited AEs after 8 days of injection** | | | | |
| Participants with solicited local AEs | | | | |
| Grade 1 | 0 | 0 | 0 | 0 |
| Grade 2 | 0 | 0 | 1 (14.3) | 0 |
| Participants with solicited systemic AEs | | | | |
| Grade 1 | 1 (10.0) | 0 | 0 | 0 |
| Grade 2 | 0 | 0 | 2 (28.6) | 0 |
| **Unsolicited TEAEs up to 28 days after Injection** | | | | |
| Participants with Treatment-Related Unsolicited TEAEs | 1 (10.0) | 1 (16.7) | 3 (42.9) | 0 |
| No. of Treatment-Related Unsolicited TEAEs | 2 | 1 | 16 | 0 |
| Participants with Any Unsolicited TEAE | | | | |
| Mild | 2 (20.0) | 3 (50.0) | 4 (57.1) | 4 (100.0) |
| Moderate | 3 (30.0) | 1 (16.7) | 2 (28.6) | 0 |

If a participant reported more than one solicited adverse events with different grades, the maximum grade was reported in this table.
*AE* adverse event, *TEAE* treatment emergent adverse event.

($n = 10$). Figure 2E shows an increase in IgG levels at D29 after a single dose of GRT-R910 administration at D1, a further increase at D142 (4 weeks following the second dose GRT-R910 dose at D113), and long-term maintenance of IgG levels for 6 months (Supplementary Dataset 1). Additional analyses assessing fold-change increases compared to baseline in cohorts receiving either 1 or 2 doses of GRT-R910 were performed to interrogate whether a second booster dose of GRT-R910 was beneficial for maintenance of $S_{D614G}$ and VOC IgG levels for at least 6 months (Supplementary Fig. S3; Supplementary Dataset 2). While the administration of a second dose of GRT-R910 did not appreciably alter fold change levels of vaccine-specific $S_{D614G}$ IgG levels, fold-change levels of cross-strain specific IgG levels to $S_{Beta}$ and $S_{Delta}$ further increased following a second GRT-R910 dose at D113. Two participants receiving a single dose of GRT-R910 (G09-101-0014 and G09-101-0020) showed reduced and delayed increases in $S_{D614G}$, $S_{Beta}$ and $S_{Delta}$, and no increase in $S_{OmicronBA1}$, IgG levels. Administration of a second GRT-R910 dose maintained higher fold change increases in $S_{OmicronBA1}$-specific IgG levels over at least 6 months compared to a single dose (Supplementary Fig. S3; Supplementary Dataset 2), suggesting increased durability of cross-binding IgG responses through 6 months following a second dose of GRT-R910. In total, $S_{WT}$ and VOCs IgG responses were significantly increased 1 month after GRT-R910 administration ($p < 0.001$, two-tailed Wilcoxon matched-pairs signed rank test) and maintained for at least 6 months following a GRT-R910 samRNA dose ($p < 0.001$, two-tailed Wilcoxon matched-pairs signed rank test; Fig. 2F; Supplementary Dataset 1).

**GRT-R910 induces cross-reactive neutralizing antibody levels to Spike VOC that are maintained for 6 months**

Neutralizing antibody (nAb) titers against live virus were assessed via microneutralization assay (MNA). Titers with 50% inhibition of the infective dose ($ID_{50}$) against SARS-CoV-2 $S_{VIC}$ were assessed for sera from participants in cohorts 1 and 2 ($n = 17$, Fig. 3A). No clear dose-response was observed when comparing $S_{VIC}$-specific neutralizing $ID_{50}$ titers between cohorts at post-vaccination timepoints (D29 and D57; Supplementary Fig. S1C). Administration of a single dose of GRT-R910 increased $S_{VIC}$ baseline nAb titers (D1: GeoMean $ID_{50} = 176.4$; Fig. 3A) ~15-fold within 4–8 weeks of GRT-R910 samRNA administration (D29: GeoMean $ID_{50} = 2,719$ and D57 GeoMean $ID_{50} = 2604$; Fig. 3A, Supplementary Dataset 1). In participants who did not receive an optional second dose of GRT-R910 ($n = 7$), $S_{VIC}$-specific neutralizing titers were maintained for 6 months after a single dose (D180: GeoMean $ID_{50} = 3194$; Fig. 3B). Similar longevity of $S_{VIC}$-specific nAb titers was

observed in participants receiving an optional second dose of GRT-R910 at D113, with $ID_{50}$ titers remaining at 1832 (GeoMean) through D293 (6 months post second dose of GRT-R910; Fig. 3C, Supplementary Dataset 1). Additional VOCs and $S_{D614G}$ were available for assessment via pseudovirus neutralization assay (PNA), and equivalence of MNA versus PNA comparing $ID_{50}$ titers against $S_{VIC}$ and $S_{D614G}$, respectively, was assessed. Values correlated significantly between the two assays (Supplementary Fig. S1D; Pearson correlation, $n = 88$, $p < 0.001$, $r = 0.6433$), confirming consistent measurement of neutralization $ID_{50}$ titers against ancestral strains $S_{VIC/D614G}$ with two different assays performed in two different laboratories. Cross-neutralization to VOC and longevity of S-specific nAb responses for 6 months following single dose of GRT-R910 was assessed via PNA for $S_{D614G}$ and VOCs Beta, Delta, and Omicron BA1 & BA5. Figure 3D shows an initial increase of $ID_{50}$ titers after a single dose of GRT-R910 administration, followed by long-term maintenance of $ID_{50}$ titers for 6 months (Supplementary Dataset 1). Longevity of Spike-specific nAb titers was also assessed in participants who received an optional second administration of GRT-R910 approximately 20 weeks (D113) post the initial GRT-R910 dose at D1 ($n = 10$). Figure 3E shows an increase in nAb levels against all S variants at D29 (4 weeks after the first dose of GRT-R910), a further increase at D142 (4 weeks post second dose of GRT-R910 at D113), and long-term maintenance of nAb levels for 6 months (Supplementary Dataset 1). Fold-change analyses comparing subjects receiving one or two doses of GRT-R910 did not reveal a clear pattern indicating a benefit of the second GRT-R910 dose (Supplementary Fig. S4, Supplementary Dataset 3). However, two participants receiving a single dose of GRT-R910 (G09-101-0014 and G09-101-0020) did not benefit from increased nAb titers against any Spike variant (Supplementary Fig. S4, Supplementary Dataset 3). In total across both dose cohorts and participants receiving either 1 or 2 doses, $S_{D614G}$ and VOC GeoMean nAb titers were significantly increased 1 month after GRT-R910 administration ($p < 0.001$, two-tailed Wilcoxon matched-pairs signed rank test) and maintained for 6 months following a GRT-R910 samRNA dose for 4 out of the 5 Spike variants assessed ($p < 0.001$, two-tailed Wilcoxon matched-pairs signed rank test; Fig. 3F, Supplementary Dataset 1).

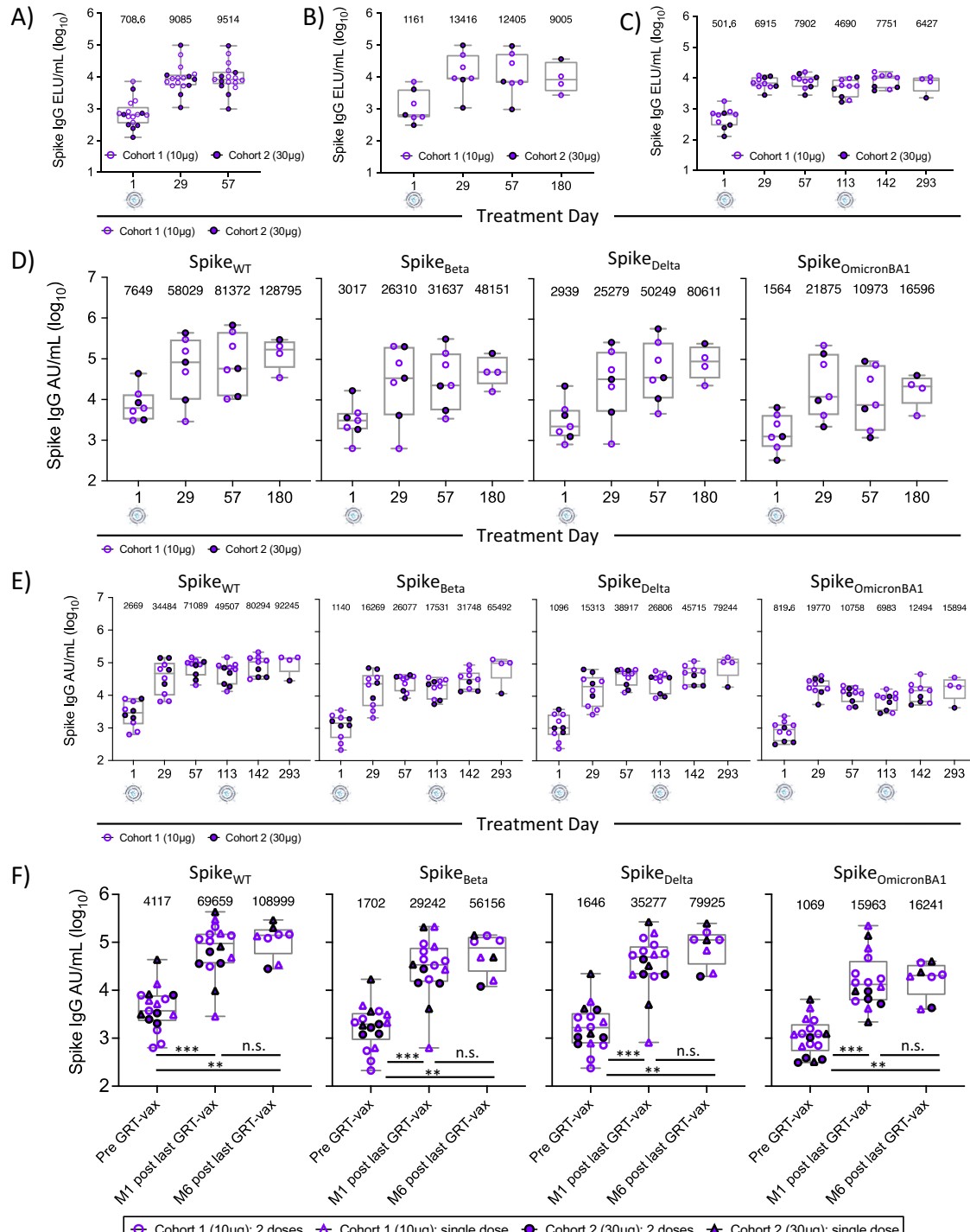

**Fig. 2 | GRT-R910 samRNA induces cross-reactive binding antibodies to Spike variants of concern (VOC) that are maintained for 6 months.** Levels of Spike$_{WT}$ and VOC Spike$_{Beta}$, Spike$_{Delta}$, and Spike$_{OmicronBA1}$ IgG assessed by ELISA (ELU/ml, **A**–**C** and/or MSD assay (arbitrary units/ml; **D**–**F** Box and Whisker plots (min-max; median; IQR) show IgG levels in relevant assay units. Geometric mean (GeoMean) for each treatment day is indicated. Individual values are shown for cohort 1 (open symbols), cohort 2 (closed symbols), single GRT-R910 dose (triangles), and two GRT-R910 doses (circles). **A** Participants in cohort 1 ($n = 10$) and cohort 2 ($n = 7$). Data from treatment day (D)1, D29, and D57. **B** Participants in cohort 1 ($n = 4$;) and cohort 2 ($n = 3$) who did not receive an optional second dose of GRT-R910 (total $n = 7$). Data from treatment days D1 ($n = 7$), D29 ($n = 7$), D57 ($n = 7$), and D180 ($n = 4$). **C** Participants in cohort 1 ($n = 6$) and cohort 2 ($n = 4$) who received an optional second dose of GRT-R910 (total $n = 10$). Data from treatment days D1 ($n = 10$), D29

($n = 10$), D57 ($n = 10$), D113 ($n = 10$), D142 ($n = 9$), and D293 ($n = 4$). **D** Participants in cohort 1 ($n = 4$) and cohort 2 ($n = 3$) who did not receive an optional second dose of GRT-R910 (total $n = 7$). Data from treatment days D1 ($n = 7$), D29 ($n = 7$), D57 ($n = 7$), and D180 ($n = 4$). **E** Participants in cohort 1 ($n = 6$) and cohort 2 ($n = 4$) who received an optional second dose of GRT-R910 (total $n = 10$). Data from treatment days D1 ($n = 10$), D29 ($n = 10$), D57 ($n = 10$), D113 ($n = 10$), D142 ($n = 9$), and D293 ($n = 4$). **F** Participants in cohorts 1 and 2 with available 6-month data following one or two GRT-R910 doses. Data from treatment days D1 ($n = 17$), 1 month (M1) post most recent GRT-R910 dose (D29 or D142, $n = 16$), and 6 months (M6) post most recent GRT-R910 dose (D180 or D293, $n = 8$). Statistical significance as assessed by two-tailed Wilcoxon matched-pairs signed rank test is indicated (n.s.: not significant, $p > 0.05$; *$p < 0.05$; **$p < 0.01$; ***$p < 0.001$).

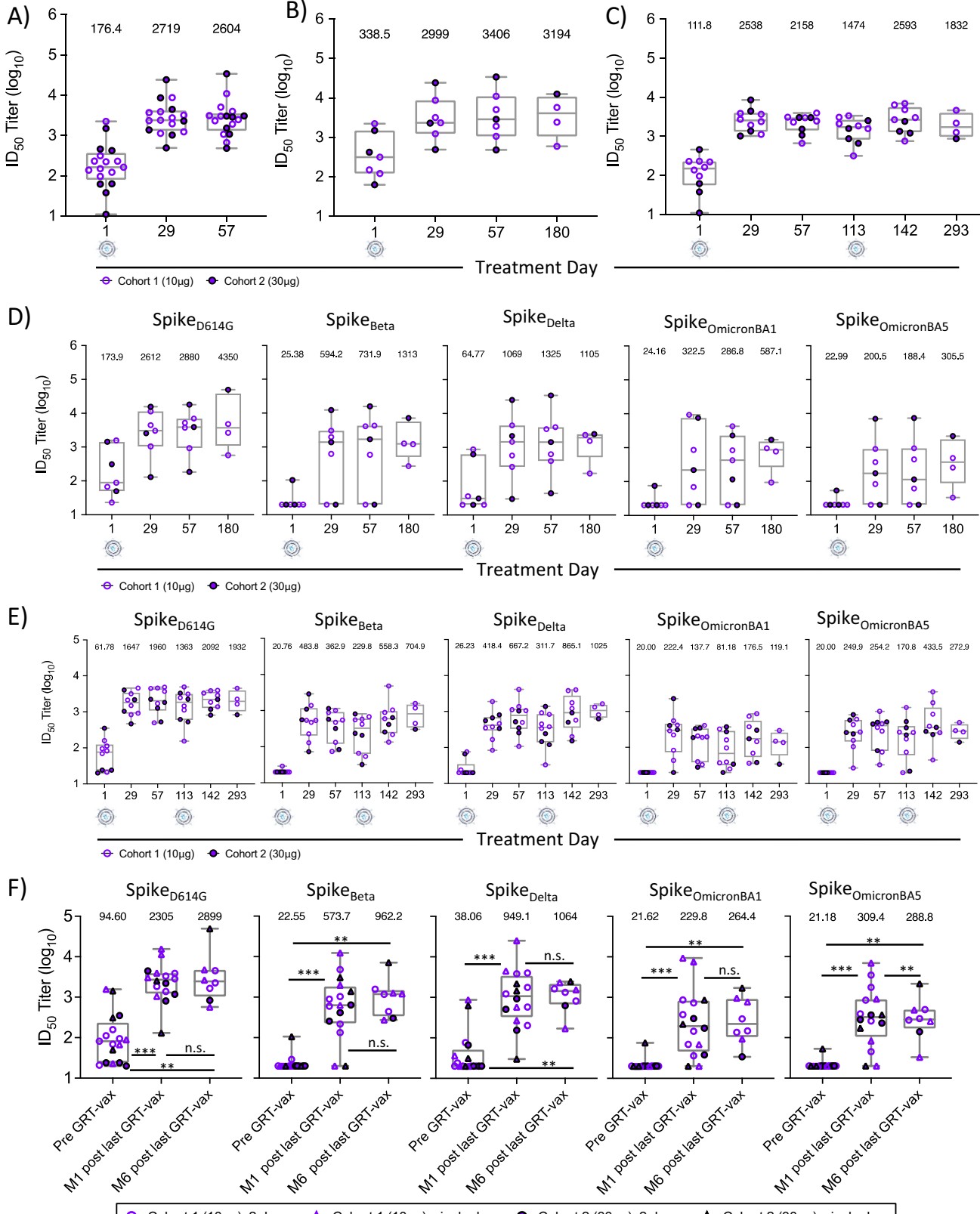

## GRT-R910 boost increases the breadth and longevity of Spike-specific T cell responses

Cellular immunity against SARS-CoV-2 $S_{D614G}$ was assessed as part of the secondary outcome measures for this interim analysis. T cell responses to SARS-CoV-2 $S_{D614G}$ were assessed via ex vivo IFN$\gamma$ ELISpot assay (methods), testing responses to overlapping peptide (OLP) pools

containing 15 amino acid peptides (15mers) spanning S subunits S1 (pools S1-1 and S1-2) and S2 (pools S2-1 and S2-2; Supplementary Dataset 4). At baseline, 12 of the 15 previously AZD1222 vaccinated participants had positive S-specific T cell responses (sum of Spike pools, GeoMean 111.4 SFU/$10^6$ cells; Fig. 4A, Supplementary Dataset 5). S-specific T cell response levels were significantly increased by D29

**Fig. 3 | GRT-R910 samRNA induces cross-reactive neutralizing antibodies to Spike VOC that are maintained for 6 months.** Levels of Spike$_{VIC}$ (Victoria/01/2020 strain; Spike$_{VIC}$), Spike$_{D614G}$, and VOC Spike$_{Beta}$, Spike$_{Delta}$, Spike$_{OmicronBA1}$, and Spike$_{OmicronBA5}$ neutralizing antibody (nAb) titers assessed by microneutralization assay (MNA; **A**–**C**) and/or pseudovirus neutralization assay (PNA; **D**–**F**). Box and Whisker plots (min-max; median; IQR) show neutralization ID$_{50}$ titers. Geometric mean (GeoMean) for each treatment day is indicated. Individual values are shown for cohort 1 (open symbols), cohort 2 (closed symbols), single GRT-R910 dose (triangles), and two GRT-R910 doses (circles). **A** Participants in cohort 1 ($n = 10$) and cohort 2 ($n = 7$). Data from D1, D29, and D57. **B** Participants in cohort 1 ($n = 4$) and cohort 2 ($n = 3$) who did not receive an optional second dose of GRT-R910 (total $n = 7$). Data from treatment days D1 ($n = 7$), D29 ($n = 7$), D57 ($n = 7$), and D180 ($n = 4$). **C** Participants in cohort 1 ($n = 6$) and cohort 2 ($n = 4$) who received an optional second dose of GRT-R910 (total $n = 10$). Data from treatment days D1 ($n = 10$), D29 ($n = 10$), D57 ($n = 10$), D113 ($n = 10$), D142 ($n = 9$), and D293 ($n = 4$). **D** Participants in cohort 1 ($n = 4$) and cohort 2 ($n = 3$) who did not receive an optional second dose of GRT-R910 (total $n = 7$). Data from treatment days D1 ($n = 7$), D29 ($n = 7$), D57 ($n = 7$), and D180 ($n = 4$). **E** Participants in cohorts 1 and 2 ($n = 4$, closed circles) who received an optional second dose of GRT-R910 (total $n = 10$). Data from treatment days D1 ($n = 10$), D29 ($n = 10$), D57 ($n = 10$), D113 ($n = 10$), D142 ($n = 9$), and D293 ($n = 4$). **F** Participants in cohorts 1 and 2 with available 6-month data following one or two GRT-R910 doses. Data from treatment days D1 ($n = 17$), 1 month (M1) post most recent GRT-R910 dose (D29 or D142, $n = 16$), and 6 months (M6) post most recent GRT-R910 dose (D180 or D293, $n = 8$). Statistical significance as assessed by two-tailed Wilcoxon matched-pairs signed rank test is indicated is indicated (n.s.: not significant, $p > 0.05$; *$p < 0.05$; **$p < 0.01$; ***$p < 0.001$).

($p < 0.05$, two-tailed Wilcoxon matched-pairs signed rank t-test; Geo-Mean 184.0 SFU/10$^6$ cells, 15/17 positive; Fig. 4A, Supplementary Dataset 5). S-specific T cell responses post administration of GRT-R910 samRNA were Th1 biased (Supplementary Fig. S5A). In participants who did not receive an optional second dose of GRT-R910 ($n = 7$), S-specific T cell responses were maintained for 6 months after a single dose (Fig. 4B, Supplementary Dataset 5). Similar longevity of S-specific T cell responses was observed in participants receiving an optional second dose of GRT-R910 at D113 (Fig. 4C, Supplementary Dataset 5). Irrespective of the number of GRT-R910 doses received, fold-change ratios comparing 1- and 6-month post-GRT-R910 boost timepoints yielded higher fold-change ratio at 6 months compared to those achieved with mRNA vaccines at 3 months in a similar boost study[27] (Supplementary Dataset 5). S-specific T cell responses were further analyzed for breadth of responses (part of exploratory outcomes measures). Figure 4D shows frequencies of responses to individual S regions represented by OLP peptide pools (Supplementary Dataset 4) at baseline (BL; 5–7 months post AZD1222 primary vaccination series) and 8 days, 29 days, and 6 months post GRT-R910 samRNA administration. Five to seven months post AZD1222 vaccination, 56.25%, 50%, 33.33% and 20% of participants had positive responses to S pools S1-1, S1-2, S2-1, and S2-2, respectively. In contrast, 6 months post GRT-R910 vaccination, 82.35%, 76.47%, 82.35% and 58.82% of participants had positive responses to S pools S1-1, S1-2, S2-1, and S2-2, respectively, showing significantly increased breadth in memory responses 6 months post-GRT compared to 5–7 months post-AZD1222 ($p < 0.05$, Chi-Square test; Fig. 4D). Overall, the frequency of participants with responses to ≥3 of the 4 S OLP pools significantly increased from 25% 5–7 months following AZD1222 vaccination to 66% 6 months after GRT-R910 vaccination ($p < 0.05$, Chi-Square test; Fig. 4E), showing either long-term boosting of previously primed and/or increasing breadth of S-specific T cell responses following GRT-R910 samRNA boost. Further exploratory analyses were performed to assess whether changes in S-specific T cell responses following GRT-R910 samRNA administration could be determined via bulk TCR sequencing. Where available, genomic DNA extracted from participants PBMC samples was analyzed for the presence of TCRb CDR3 sequences specific for identified SARS-CoV-2 T cell epitopes via bulk DNA immunosequencing (ImmunoSEQ). Analysis of TCRb CDR3 sequences revealed increased breadth and depth of putative TCR clonotypes recognizing S sequences following GRT-R910 samRNA administration compared to baseline (Fig. 4F), corroborating functional data showing increased magnitude and breadth of S-specific functional T cell responses.

## T cell responses to TCE components following GRT-R910 administration are broad and maintained for 6 months

T cell responses to SARS-CoV-2 TCE regions (Fig. 1A) were assessed via post in vitro stimulation (IVS) IFN$\gamma$ ELISpot assay, testing responses to OLP pools spanning TCE regions of Nucleocapsid (N), Membrane (M) and ORF3a (Supplementary Dataset 4). T cell responses to TCE sequences were evaluated in healthy donor samples collected prior to 2020 (pre-pandemic donors) to assess whether conserved TCE regions included in GRT-R910 were able to stimulate pan-coronavirus T cell responses. While most pre-pandemic donors did not have detectable T cell responses to TCE via ex vivo ELISpot, 6/6 and 5/6 donors tested had positive responses to TCE N and ORF3a epitopes, respectively, after brief antigen-specific expansion (Supplementary Fig. S5B, C), confirming pan-coronavirus epitopes in GRT-R910. Administration of a single dose of GRT-R910 samRNA increased primarily ORF3a-specific T cell responses, though increases were not statistically significant ($p > 0.05$, two-tailed Wilcoxon matched-pairs signed rank t-test; Fig. 5A). T cell responses to TCE components did not decrease over 6 months after a GRT-R910 dose (Supplementary Fig. S6A, B, Supplementary Dataset 5). Fold-change ratios of Nuc, Mem, and ORF3a-specific T cell responses 1 month (M1) and 6 months (M6) post GRT-R910 administration suggest higher levels of memory T cell populations 6 months following a single dose of GRT-R910, rather than following two doses (Supplementary Fig. S6 and Dataset 5). Pre-existing responses to TCE regions Nuc, Mem, and ORF3a were detected at BL in 10/15, 3/15, and 9/15 participants, respectively, and maintained following GRT-R910 administration (Fig. 5B). Importantly, in participants with no pre-existing T cell responses to Nuc (5/15), Mem (12/15), and ORF3a (6/15), de novo T cell responses to Nuc, Mem and ORF3a were primed in 3/5, 5/12, and 5/6 participants following GRT-R910 administration, respectively (Fig. 5B). Frequencies of T cell responses to individual TCE regions represented by OLP or minimal peptide pools (Supplementary Dataset 4) were assessed. Five to seven months after AZD1222 vaccination (BL) 66.7%, 20%, 60%, 16.7%, and 18.2% of participants had positive responses to TCE pools Nuc OLP, Mem OLP, ORF3a OLP, Nuc Min, and ORF3a Min, respectively (Fig. 5C). Comparatively, 100%, 42.9%, 85.7%, 25%, and 28.6% of participants had positive responses 6 months following GRT-R910 administration to TCE pools Nuc OLP, Mem OLP, ORF3a OLP, Nuc Min, and ORF3a Min, respectively, though increases did not reach statistical significance ($p > 0.05$, Chi-Square test; Fig. 5C). Overall, the frequency of participants with responses to ≥3 of the 5 TCE OLP and minimal peptide pools significantly increased from 24% at baseline to 62% at 6 months ($p < 0.0001$, Chi-Square test; Fig. 5D), showing increasing and maintained breadth of TCE-specific T cell responses following the GRT-R910 samRNA administration. Analysis of TCRb CDR3 sequences revealed increased breadth and depth of putative TCR clonotypes recognizing TCE regions following GRT-R910 samRNA administration compared to baseline (Fig. 5E), corroborating functional data showing priming of de novo responses, as well as increased magnitude and breadth of TCE-specific functional T cell responses.

## GRT-R910 administration induces polyfunctional CD4$^+$ and CD8$^+$ T cells specific to non-Spike epitopes included in the vaccine

Functional profiling of SARS-CoV-2 specific CD4$^+$ and CD8$^+$ T cells was assessed by intracellular cytokine staining (ICS) as part of exploratory outcome measures. CD4$^+$ and CD8$^+$ T cell responses to TCE

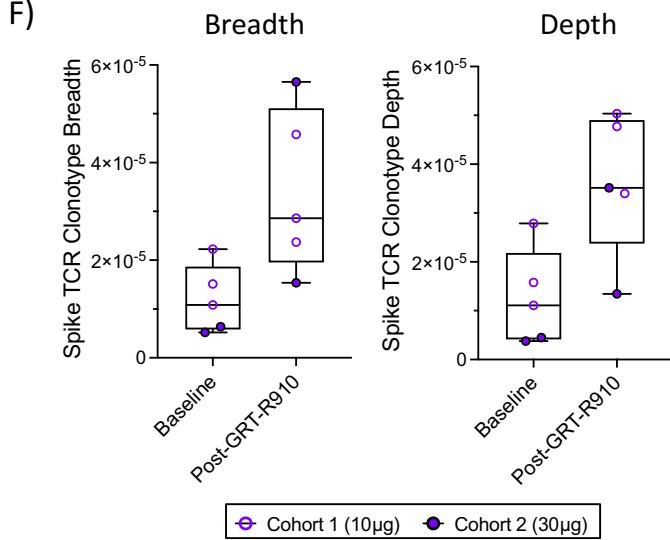

**Fig. 4 | GRT-R910 administration induces long-lived Spike-specific T cells and TCR repertoires with increased breadth and depth.** T cell responses to overlapping peptide pools assessed by ex vivo IFNγ ELISpot assay. Box and Whisker plots (min-max; median; IQR) show spot forming units per million PBMCs (SFU/$10^6$). GeoMean for each treatment day is indicated. Circles indicate positive (>LOD or >2x DMSO) responses, triangles indicate negative (<LOD or <2x DMSO) responses. Open and closed symbols indicate cohorts 1 and 2, respectively. D1 samples not available for participants 0003 and 0004 due to poor PBMC sample quality. **A** T cell responses to Spike for cohorts 1 ($n = 10$) and 2 ($n = 7$). Data from D1 ($n = 15$), D8 ($n = 17$), and D29 ($n = 17$). Statistical significance as assessed by two-tailed Wilcoxon matched-pairs signed rank test comparing D1 versus D29 is indicated (*$p = 0.0103$). **B** T cell responses to Spike for cohorts 1 ($n = 4$) and 2 ($n = 3$) without optional second dose of GRT-R910 (total $n = 7$). Data from D1 ($n = 7$), D8 ($n = 7$), D29 ($n = 7$), and D180 ($n = 4$). **C** T cell responses to Spike for cohorts 1 ($n = 6$) and 2 ($n = 4$) with optional second dose of GRT-R910 (total $n = 10$). Data from D1 ($n = 10$), D8 ($n = 10$), D29 ($n = 10$), D113 ($n = 10$), D142 ($n = 9$), and D293 ($n = 4$). **D** Percent (%) of participants ($n = 17$) with positive responses to Spike OLP pools S1-1 (turquoise), S1-2 (light blue), S2-1 (medium blue), and S2-2 (navy blue) at D1, D8, D29, and 6 months are shown. Statistical significance as assessed by two-sided Chi-square test comparing D1 versus 6 months post GRT-R910 dose (M6) is indicated (*$p = 0.0275$). **E** Pie charts depicting frequency (%) of participants ($n = 17$) positive for number of Spike OLP pools (0 = gray; 1 = lavender; 2 = light purple; 3 = medium purple; 4 = dark purple) at D1, D8, D29, and M6. Percentages for each pie slice are indicated. **F** Spike-specific TCRb CDR3 clonotype sequence breadth and depth are shown for a subset of participants ($n = 5$; $n = 3$ from cohort 1; $n = 2$ from cohort 2). Box and Whisker plots (min-max; median; IQR) show TCRb CDR3 clonotype sequence breadth and depth for D1 and post administration of GRT-R910.

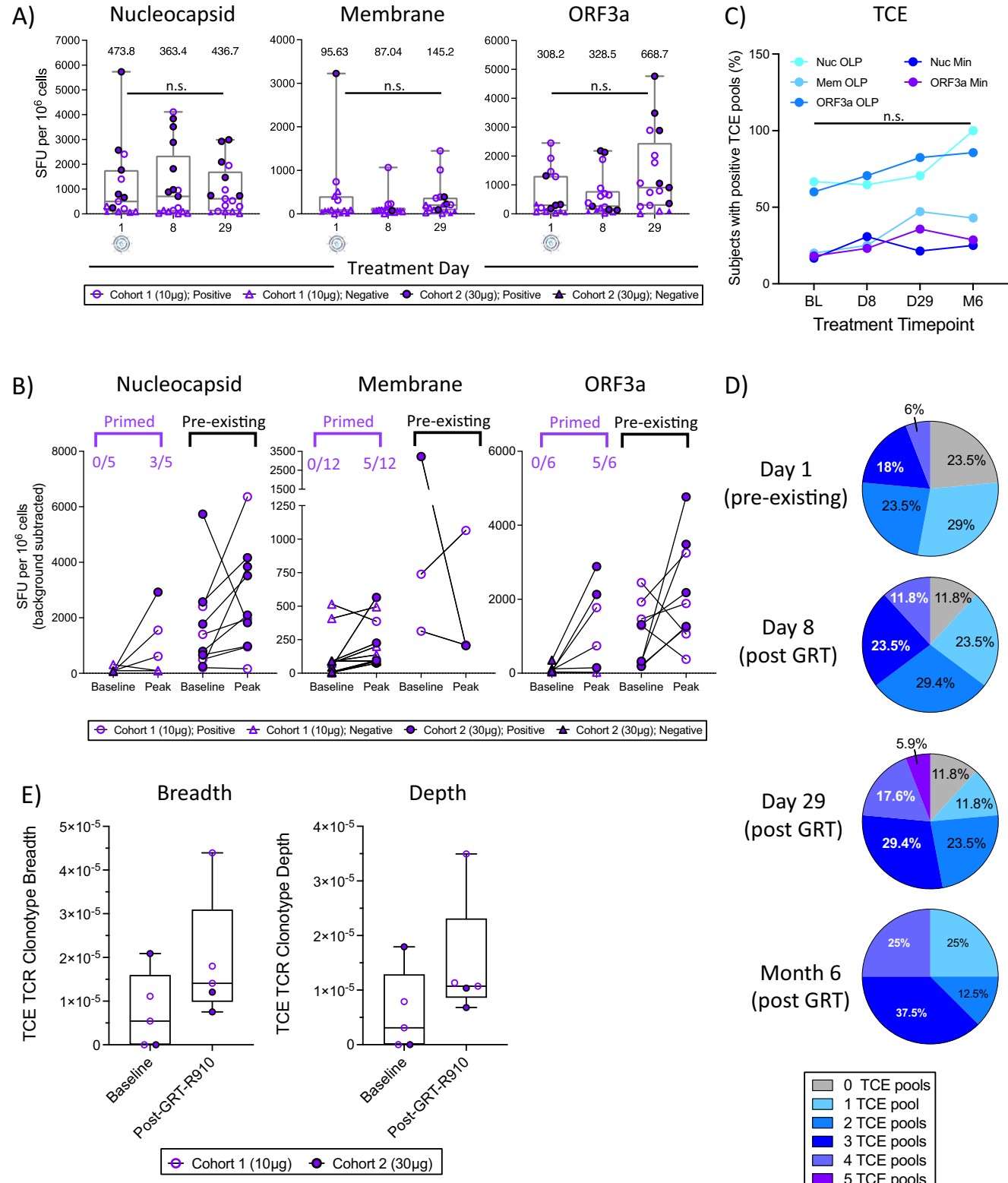

components N, ORF3a, and M were assessed via stimulation with OLP pools or pools consisting of predicted HLA-restricted minimal epitopes contained within ORF3a or N (TCE OLP; ORF3a OLP; Nuc OLP; Mem OLP, ORF3a Min; Nuc Min; Supplementary Dataset 4). Responses to minimal epitope pools were tested ad hoc where sufficient cellular material was available, as HLA-typing information was not available for participants enrolled in this study. Figure 6A shows representative data from post-treatment timepoint PBMCs from an individual with ORF3a

responses stimulated with TCE and ORF3a OLP pools, as well as ORF3a Min pool. CD8+ T cells stimulated with ORF3a minimal epitopes showed increased IFNγ production (7.19% CD8+IFNγ+ cells) compared to either vehicle control, total TCE OLP, or ORF3a OLP stimulated conditions (2.39%, 3.72%, and 3.96% CD8+IFNγ+ cells, respectively). In contrast, CD4+ T cells from the same individual showed increased IFNγ production (6.12% CD4+IFNγ+ cells) when stimulated with ORF3a OLP pools compared to either vehicle, TCE OLP, or ORF3a Min stimulation

**Fig. 5 | T cell responses to epitopes within TCE following GRT-R910 samRNA administration are broad and maintained for 6 months.** T cell responses to TCE regions assessed by post-IVS ELISpot. Box and Whisker plots (min-max; median; IQR) show background-subtracted spot forming units per million PBMCs (SFU/10⁶). GeoMean for each treatment day is indicated. Circles indicate positive (>LOD or >2x DMSO) responses, triangles indicate negative (<LOD or <2x DMSO) responses. Open and closed symbols indicate cohorts 1 and 2, respectively. D1 samples not available for participants 0003 and 0004 due to poor PBMC sample quality. **A** T cell responses to TCE components Nucleocapsid, Membrane, and ORF3a are shown for cohorts 1 (n = 10) and 2 (n = 7) for D1 (n = 15), D8 (n = 17), and D29 (n = 17). Statistical significance was assessed by two-tailed Wilcoxon matched-pairs signed rank test comparing D1 versus D29 (n.s.: not significant, p > 0.05). **B** Before-after dot plots showing individual responses in participants with (Nuc n = 10; Mem n = 3; ORF3A n = 9) or without (Nuc n = 5; Mem n = 12, ORF3a n = 6) pre-existing responses to TCE

components are shown (post-IVS at D1 versus subsequent peak responses post GRT-R910 dose). Fraction of primed responses indicated. **C** Percent (%) of participants with positive responses to Nucleocapsid (turquoise), Membrane (light blue), ORF3a (blue), Nucleocapsid minimal epitopes (navy blue), and ORF3a minimal epitopes (purple) at D1 (n = 15), D8 (n = 17), D29 (n = 17), and 6 months (M6; n = 8) are shown. Statistical significance as assessed by two-sided Chi-square test comparing BL versus M6 is indicated (n.s.: not significant, p > 0.05). **D** Pie charts depicting frequency (%) of participants (n = 17) positive via post-IVS ELISpot for number of TCE pools (0 = gray; 1 = light blue; 2 = blue; 3 = navy blue; 4 = light purple; 5 = dark purple) at treatment days 1, 8, 29, and 6 months. Percentages for each pie slice are indicated. **E** TCE-specific TCRb CDR3 clonotype sequence breadth and depth are shown for a subset of participants (n = 5; n = 3 from cohort 1; n = 2 from cohort 2). Box and Whisker plots (min-max; median; IQR) show TCRb CDR3 clonotype sequence breadth and depth for D1 and post administration of GRT-R910.

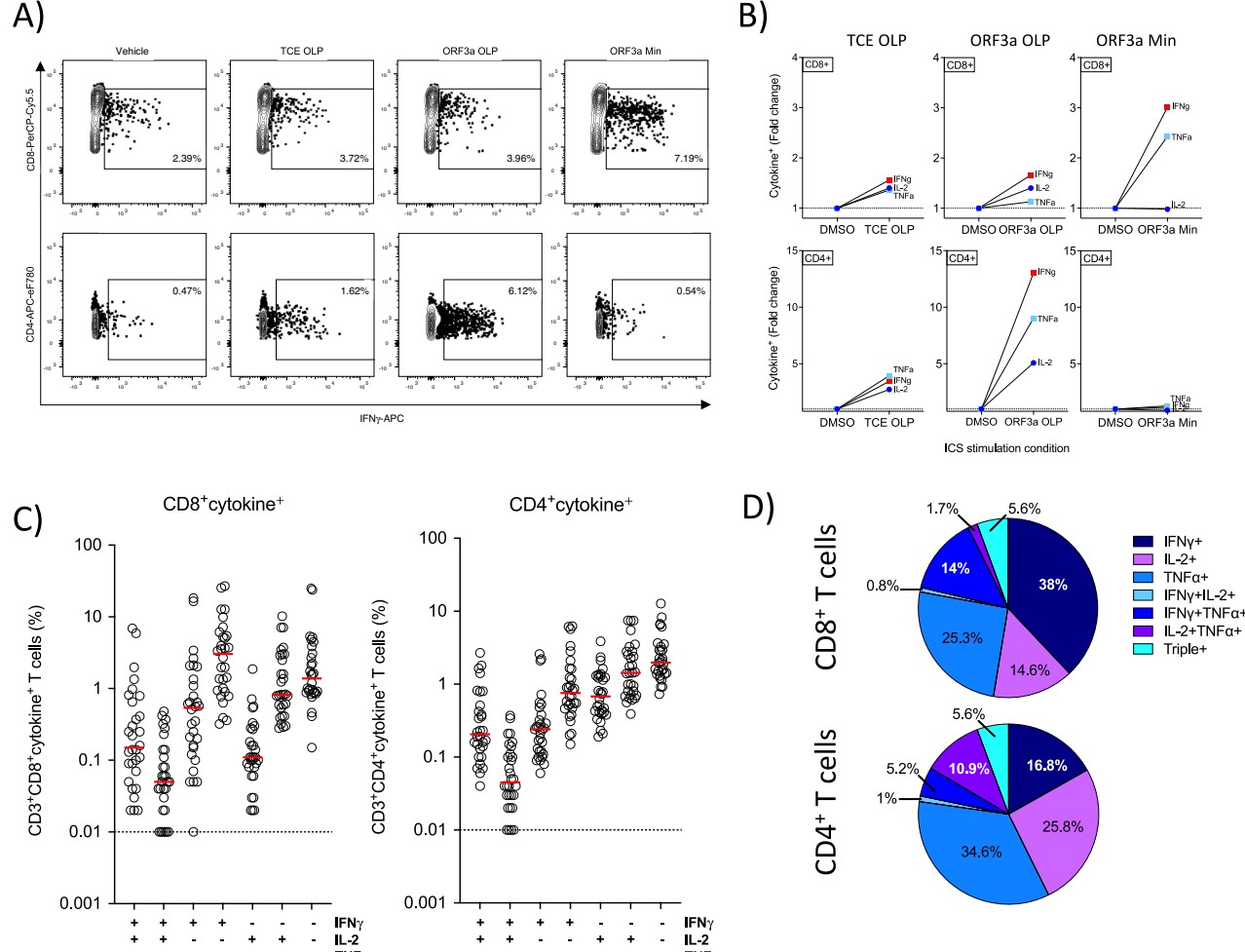

**Fig. 6 | GRT-R910 samRNA administration induces polyfunctional CD4⁺ and CD8⁺ T cells specific to non-Spike epitopes included in the TCE part of the vaccine.** Cytokine production to TCE components was assessed via intracellular cytokine staining (ICS) for CD4⁺ and CD8⁺ T cells. **A** Representative dot plots for participant GO9-101-0003 (cohort 1) showing IVS-expanded CD8⁺ (top row) and CD4⁺ (bottom row) T cells producing IFNγ following overnight stimulation with DMSO (vehicle), TCE OLP pool, ORF3a OLP pool, or ORF3a minimal epitope pool (Supplementary Dataset 4). Cells were gated on PBMCs (FSC-A vs SSC-A), singlets (FSC-A vs FSC-H), live cells (FSC-A vs Live/Dead), CD3⁺ cells, and CD8⁺ or CD4⁺ T cells and IFNγ⁺ cells. **B** Fold change IFNγ (red square), TNFα (light blue square),

and IL-2 (dark blue circle) expression for CD8⁺ or CD4⁺ T cell (top and bottom graphs, respectively) stimulated with TCE OLP pool, ORF3a OLP pool, or ORF3a minimal epitope pool relative to vehicle-stimulated cells for participant GO9-101-0003. **C** Polyfunctionality was assessed via Boolean gating of CD8⁺cytokine⁺ populations (background subtracted). Scatter dot plots with median indicated (red line) show frequencies of TCE-specific cytokine positive CD8⁺ or CD4⁺ T cells following GRT-R910 samRNA administration are shown for single-, double-, and triple-positive populations (n = 30). **D** Pie charts depict frequency of single-, double-, and triple-positive CD8⁺ and CD4⁺ T cell populations after GRT-R910 administration.

conditions (0.47%, 1.62%, and 0.54% CD4$^+$IFNγ$^+$ cells for vehicle, respectively). CD8$^+$ T cell driven IFNγ and TNFα responses to ORF3a minimal epitopes in this individual were increased 3-fold and 2.5-fold, respectively, while no significant changes in IL-2 production were observed (Fig. 6B). In comparison, CD4$^+$ T cell driven IFNγ, TNFα, and IL-2 responses increased 13-fold, 3-fold, and 5-fold, respectively, when stimulated with ORF3a OLP pools (Fig. 6B). Polyfunctionality and frequencies of single-, double-, and triple-positive CD4$^+$ and CD8$^+$ T cell responses to TCE peptide pools were assessed (Fig. 6C, D). Overall, CD8$^+$ T cell responses were driven by IFNγ and TNFα production (77.3% of response; median 3.02% IFNγ$^+$; 1.38% TNFα$^+$, and 0.535% IFNγ$^+$TNFα$^+$ cells) with less IL-2 production (median 0.82% IL-2$^+$ cells), while CD4$^+$ T cell responses were driven by TNFα and IL-2 responses (71.3% of response, median 1.965% TNFα$^+$, 1.43% IL-2+, 0.68% TNFα$^+$IL-2$^+$ cells), with lower IFNγ responses (median 0.755% IFNγ$^+$ cells).

## Discussion

This is the first clinical trial to assess the safety and immunogenicity of a samRNA-based SARS-CoV-2 vaccine candidate GRT-R910 delivering Spike and T cell epitope sequences from conserved genes outside of Spike in healthy adults ≥60 years of age following a prior AZD1222 vaccine primary series. In the dose-escalation part of the trial described in this interim analysis, participants received a single dose of GRT-R910 (10 or 30 μg) followed by an optional second dose approximately 4 months later. Safety and tolerability were assessed as primary outcome measures. When administered as a one- or two-dose regimen after a primary series of AZD1222, GRT-R910 was well-tolerated. Treatment-related adverse events were consistent with a vaccine-induced immune response, mostly Grade 1 or 2 in severity, and transient in nature. Participants in the 30 μg cohort experienced moderately increased severity and duration of solicited adverse events. Overall, the safety and tolerability profile of GRT-R910 was similar to that observed following primary vaccinations with mRNA vaccines in younger and older adults[28,29], as well as that observed in adults receiving an mRNA boost following prior AZD1222 vaccine primary series[30]. In line with a recent report about another self-amplifying mRNA based SARS-CoV-2 vaccine administered as primary series in adults[31], AEs observed with GRT-R910 did not increase in number and severity with administration of a second dose. These observed differences between AEs described post samRNA or mRNA vaccinations could be due to use of different doses (lower samRNA doses versus higher mRNA doses) or differences in LNP composition and/or innate immune responses to mRNA versus samRNA[32]. Differences in dosing intervals likely did not contribute to the difference in AEs observed post the second dose of mRNA1273 compared to GRT-R910, since monthly samRNA boosts for up to a year did not result in an increase in number and severity of AEs in oncology studies[19].

In line with boosting effects observed for authorized mRNA vaccines following mRNA- or Adenovirus-based primary series[27,30], GRT-R910 substantially boosted both binding and neutralizing antibodies to S in both dose groups. Notably, a 10 μg dose of GRT-R910 boosted IgG and nAb responses to similar levels as those observed following a 100 μg boost of mRNA-1273 in another study with similar design[30]. The most notable differentiator between mRNA and samRNA vaccine platforms is the replication of samRNA within the transduced cells post vaccine administration, driving prolonged antigen expression, increased durability of nAb titers, and offering dose sparing potential in mice[22,23]. NHP studies have shown that the same samRNA-based vaccine platform as the one used for GRT-R910 induced potent nAb titers and provided protection from SARS-CoV-2 infection post-vaccination at a 10 μg dose, similar to levels of protection observed in NHPs with 10-fold higher doses (100 μg) of authorized SARS-CoV-2 mRNA vaccines[20,33,34]. Analogous to observations in younger adults vaccinated with increasing doses of a self-amplifying mRNA vaccine[35], administration of the 30 μg dose of GRT-

R910 samRNA did not significantly increase IgG or nAb levels compared to the 10 μg dose. This suggests that optimal samRNA replication and antigen expression levels for boosting of humoral responses were achieved with the lower dose of 10 μg, indicating highly desirable dose-sparing properties of GRT-R910 samRNA. Interestingly, though fold-change analyses suggested that administration of a second dose of GRT-R910 was beneficial with regards to increasing cross-strain specific IgG levels, this was not apparent in the nAb data. However, it is possible that administration of a second dose of GRT-R910 could boost antibody levels in individuals with stronger immune senescence that might not benefit as much from a single dose. Currently authorized vaccines played an important and successful role in reducing the mortality and morbidity associated with COVID-19[3,4,36], but the rapid dissemination of new VOC[36,37] and relatively short durability in humoral response of authorized vaccines[5,38–42] necessitates continued investigation into next generation vaccines that provide durable and broad cross-variant protection. Accordingly, 3-month follow up data from the COV-BOOST study showed declining antibody levels at D84 compared to D28 in subjects receiving an mRNA booster dose (BNT and m1273) in subjects who had received an adenoviral-based primary vaccination series[27]. Data from a recent study assessing safety and immunogenicity of a self-amplifying mRNA (ARCT-021) administered as a one- or two-dose primary series in younger (21–55 years of age) and older (56–80 years of age) adults showed a marked decline in nAb levels 3 months following vaccination, irrespective of the age group[31]. In addition to boosting IgG and nAb levels to the vaccine-relevant strain (S$_{D614G}$), GRT-R910 boosted IgG and nAb responses to the evaluated VOC. Importantly, while 1/17 (6%) of dosed participants were seropositive for nAb responses to VOC Omicron BA1 and BA4/5 19–31 weeks following AZD1222 primary series, administration of a single 10 μg dose of GRT-R910 samRNA resulted in 90% seroconversion rate of nAb responses against VOC Omicron BA1 and BA4/5 (9/10 > LOD at D29). Although direct comparisons are not appropriate as ARCT-021 was administered as a primary series and GRT-R910 was administered as a boost following an AZD1222 primary series, it is interesting to note that nAb titers induced by ARCT-021 in individuals with no detectable baseline titers waned within 3 months of administration (ARCT-021 D85 GeoMean titers ranging from 20 (LOD) to 27)[31]. In comparison, cross-reactive nAb titers to VOC Omicron BA1/BA5 induced in seroconverted subjects did not decrease (BA1), or were maintained above LOD (BA5), 6 months following GRT-R910 administration. Differences in longevity of antibody responses could be attributed to the difference between de novo induction versus boosting, but could also be due to inherent differences in the LNP used for samRNA formulation, differences in Spike sequence and/or antigen cassette design, as well as differences in the samRNA backbone between GRT-R910 and ARCT-021[20,23]. One major limitation of the data presented in this interim analysis is the small sample size, and potential cross-strain humoral immunity and longevity of response boosted by GRT-R910 needs to be corroborated by larger datasets. Additional data from this study, as well as a study assessing longevity of antibody responses induced by similar samRNA constructs (GRT-R912, GRT-R914, and GRT-R918) as primary vaccination series in an ongoing Phase 1 study in South Africa in SARS-CoV-2 exposed and unexposed individuals (NCT05435027) will be required to corroborate or disprove preliminary finings presented in this interim report.

While humoral immunity induced by authorized vaccines wanes within 6 months[5], S$_{D614G}$ and VOC-specific IgG and nAb levels induced or increased by GRT-R910 were maintained following the most recent GRT-R910 dose, with no decline in IgG concentrations or nAb titers for at least 6 months in participants with evaluable long-term timepoints. These data suggest that GRT-R910 may be able to induce long-lasting humoral immunogenicity (currently assessed up to

6 months) against multiple variants, including ones that were not included in the vaccine. While based on a very small sample set, if confirmed in ongoing cohorts and other studies assessing similar samRNA constructs, these preliminary data hold promise for significantly increased durability of vaccine-induced humoral responses against VOC with samRNA at relatively low doses compared to authorized mRNA vaccines. This is particularly important in this older population since COVID-19 vaccines are less immunogenic in older compared to younger adults, which is attributed to age-related immunosenescence[43]. Both dose-sparing and induction of long-lived immune responses following vaccination with samRNA compared to mRNA can likely be attributed to difference in antigen expression kinetics and corresponding prolonged immune priming in samRNA vs mRNA[44].

Several lines of evidence support the importance of T cell responses in protecting from severe SARS-CoV-2 disease[15–17,45], and T cell responses following natural infection display immunodominance patterns, with both CD4[+] and CD8[+] T cell responses focusing on dominant epitopes in S, M, and N, and, to a lesser extent, ORF3a[46]. GRT-R910 was designed to induce antibody responses against S (expressed as whole protein in a prefusion format), as well as T cell responses to S and highly conserved non-Spike T cell epitope sequences in N, ORF3a, and M using a previously published prediction algorithm[24] and selected validated epitopes[25,26]. GRT-R910 samRNA boosted S-specific T cell responses primed by AZD1222 primary series[47], with significantly increased breadth of S-specific T cell responses evident 6 months after GRT-R910 administration compared to responses observed 5–7 months after AZD1222 administration. In addition, GRT-R910 primed de novo T cell responses to non-S SARS-CoV-2 epitopes from N, ORF3a, and M and broadened the response across epitopes. The generation and maintenance of antigen-specific memory T cells is crucial for long-term effective vaccination and immune protection, but there are important functional changes in both naïve and memory populations in older adults[48]. Even with a limited sample size, our early data suggest that GRT-R910 is able to produce a long-term immune response in this older population. If confirmed, GRT-R910 could theoretically be administered less frequently compared to first generation COVID-19 boosters while providing comparable, or superior, protection from severe COVID-19. Although participants on this study were at higher risk of severe SARS-CoV-2 infection due to their age group, those infected following GRT-R910 administration experienced mild SARS-CoV-2 symptoms and did not require hospitalization. This observation is limited to a small sample size, and additional data are required to support any conclusions on whether GRT-R910-induced immunity provides protection from severe disease in more vulnerable populations. Studies assessing SARS-CoV-2 epitopes constrained by structural and networked positions highlight that vaccine designs should focus on including epitopes in addition to S[49], with a recent study showing induction of strong T cell responses to selected non-S epitopes in the absence of antibody induction in an adjuvanted peptide-based vaccine[50]. However, vaccines that exclusively induce T cell responses in the absence of induction of humoral immunity are unlikely to provide sufficient protection from current and emerging SARS-CoV2 VOCs, especially if the exposure dose to the virus is high, as both humoral and cellular immunity are important in reducing infection rates and disease severity[3,4,16,17]. This interim analysis provides preliminary evidence that a samRNA-based vaccine (GRT-R910) targeting both full-length Spike and conserved non-Spike T cell epitopes, at doses as low as 10 μg, induces and boosts broad T cell responses in addition to durable, cross-neutralizing antibodies that are maintained for at least 6 months, suggesting that samRNA might offer the opportunity of a dose-sparing vaccine platform that is able to drive durable cellular and humoral immunity against viruses and other pathogens.

## Methods

### Ethics statement
The trial was conducted in accordance with the ethical principles derived from international guidelines, including the Declaration of Helsinki (7th revision, 2013), the international ethical guidelines of the Council for International Organizations of Medical Sciences, applicable Good Clinical Practice guidelines of the International Council for Harmonization, and all applicable laws and regulations. The trial protocol and all other relevant documentation were reviewed and approved by a local or central institutional review board or ethics committee for each site. All the participants provided written informed consent (with assistance from a legally authorized representative if required) before enrollment. All 3 sites on the study use the NSH Health Research Authority, London – West London & GTAC Research Ethics Committee. The sites are University Hospitals Birmingham NHS (Birmingham, United Kingdom), University Hospital of Leicester NHS Trust (Leicester, United Kingdom), Manchester University (Manchester, United Kingdom).

### Study design
This on-going phase 1 multi-center, open label, dose-escalation trial was designed to examine dose, safety, tolerability (primary endpoints), and immunogenicity (secondary endpoint) of GRT-R910, an investigational samRNA SARS-CoV-2 vaccine when administered to healthy adults 18 years and older who were previously vaccinated with a first-generation SARS-CoV-2 vaccine, AZD1222 or an mRNA-based SARS-CoV-2 vaccine (Fig. 1a). Date of preregistration at EUDRACT: 21/Aug/2020 and study details are listed at https://clinicaltrials.gov/ct2/show/NCT05148962?term=gritstone&draw=2&rank=3. All participants have provided written informed consent prior to initiation of any study procedures. The trial is being conducted in accordance with the International Council for Harmonization of Technical Requirements for Pharmaceuticals for Human Use, Good Clinical Practice guidelines, and applicable government regulations. The UK national institutional review board (IRB) approved the protocol and the consent forms. This report presents outcomes in Cohorts 1 and 2, which enrolled healthy adults ≥60 years of age who were previously vaccinated with a primary series of AZD1222. This study is also assessing the safety and immunogenicity of GRT-R910 as a two-dose regimen (1-month interval) administered to healthy adults 18 years and older who were previously vaccinated with an adenoviral COVID-19 vaccine or a mRNA-based COVID-19 vaccine in separate cohorts. Participants were not compensated for participating in this study.

### Study population and cohort assignment
Eligible participants for cohorts 1 and 2 were ≥60 years of age who have received AZD1222 vaccine as part of a clinical trial or under the National Deployed Vaccine Programme in the UK (Fig. 1B). All participants were required to have completed their two-dose regimen of their primary vaccination series at least 2 months prior to study entry. Participants who had a history of prior confirmed SARS-CoV-2 infection or active infection were excluded from the study. The study design doesn't take into account sex/gender since it is not expected to see differences between the sex/gender with respect to the toxicity and tolerability within each small cohort of 10 subjects. Sentinel dosing in cohorts 1 and 2 consisted of two participants in each cohort dosed 72 h ahead of the remainder of each cohort, with the remainder of the participants dosed at the same dose level only if no halting rules were met. Enrollment in cohort 2 proceeded after safety data from all participants in cohort 1 met the dose-escalation rules. The reactogenicity and immunogenicity data assessed after the first dose of GRT-R910 informed selection of the 10 μg dose of GRT-R910 currently being evaluated in the remaining ongoing cohorts. Initially, the GO-009 study was designed for participants to receive a single dose of GRT-R910. In order to evaluate the benefit of a two-dose regimen (to

evaluate the prime and boost effect of the TCE component of the vaccine candidate), participants were subsequently offered an optional second dose of GRT-R910, ~4 months after the first dose of GRT-R910.

## Data collection

Clinical for this interim analysis data were collected in an electronic data capture system database based on case report forms. Solicited AEs and unsolicited TEAEs are summarized as counts and percentages, and any AEs after the administration of GRT-R910 vaccination were coded by Medical Dictionary for Regulatory Activities (MedDRA) version 25.0. Enrollment for cohorts 1 and 2 opened in September 2021 and was completed in December 2021. Immunogenicity data through 6 months following last dose of GRT-R910 were generated and analyzed by the study sponsor or VisMederi Srl. (Sienna, Italy) using participants' serum or peripheral blood mononuclear cells.

## Safety assessment

Safety assessments included collecting local and systemic solicited adverse events (AEs) from the time of GRT-R910 administration within 7 days (Day 1 through Day 8) after each GRT-R910 administration. Unsolicited treatment-emergent adverse events (TEAEs) were captured for 28 days, and serious AEs (SAEs) and adverse events of special interests (AESIs) are recorded for 1 year following each GRT-R910 vaccination. All AEs were assessed for severity, according to the toxicity grading scales in the FDA guidance document entitled "Toxicity Grading Scale for Healthy Adult and Adolescent participants Enrolled in Preventive Vaccine Clinical Trials".

## Statistical methods

The primary outcomes of this study are reactogenicity, local and systemic solicited AEs, and immunogenicity. Only descriptive statistics are used for data analyses since the sample size of 10 in each cohort was not powered for hypothesis testing. Solicited AEs and unsolicited TEAEs are summarized as counts and percentages, and any AEs after the administration of GRT-R910 vaccination were coded by Medical Dictionary for Regulatory Activities (MedDRA) version 25.0. Immunogenicity data of SARS-CoV-2 neutralizing antibody and spike IgG levels against wild type, Beta, Delta, and Omicron (BA1, BA4 and BA5) variants are summarized using geometric mean (GeoMean) titers and presented in box plots showing minimum, maximum, outlier, and median values. Immunogenicity data for SARS-CoV-2 T cell responses are shown as spot forming units (SFU) per million cells. Since GeoMean values cannot be calculated for datasets including zeros, ELISpot values equal to zero resulting from ex vivo and post-IVS ELISpot assays have been set to ½ the LOD for the relevant assay (ex vivo ELISpot LOD = 30 SFU/$10^6$ cells; post-IVS ELISpot LOD = 180 SFU/$10^6$ cells) to enable calculation of GeoMean values. The data cut off was 23 August 2022. Statistical analyses of immune data were performed using SAS/STAT® Software Version 9.4 for PC (SAS, Cary, NC, USA) and GraphPad Prism Version 9.0.1 for macOS, and included Shapiro-Wilk test to assess parametric versus non-parametric data distributions, two-tailed Wilcoxon matched-pairs signed rank test (Figs. 2F, 3F, 4A, 5A), Chi-Square test (Figs. 4D, E, 5C, D), two-tailed Mann-Whitney test (Figs. S1A, S1C), and Pearson Correlation (Supplementary Figs. S1B; S1D).

## T cell epitope (TCE) cassette design

Genes of interest were selected based on published data[25,26], selecting epitopes with a length of 8-11 amino acids from S, N, M, and ORF3a after predicting their presentation across all available alleles in our presentation model[24]. Sequencing data for over 80,000 SARS-CoV-2 isolates were collected from GISAID[51] and identified mutation sites, removing any epitope that spanned a mutation site with greater than 1% frequency. Remaining epitopes were annotated as validated in the literature on SARS-CoV-2 or conserved SARS-CoV-1 epitopes. Cassette

design was initialized with two validated epitopes in M and all validated and predicted epitopes in N and ORF3a. Adjacent epitopes in frames covering both primary and flanking sequences of epitopes were added such that frames were no more than 60 amino acids in length, organized to minimize redundant overlapping sequence content, terming the total size of all frames our TCE footprint. Sets of adjacent amino acids were iteratively removed from N and ORF3a to reduce footprint size while maintaining population HLA haplotype coverage loss[52].

## GRT-R910

GRT-R910 (Fig. 1A) is a samRNA vector based on the Venezuelan Equine Encephalitis Virus (VEEV) formulated into lipid nanoparticles (LNP). To generate GRT-R910, sequences encoding the structural proteins of VEEV were deleted and replaced by an expression cassette encoding full-length Spike ($S_{D614G}$) protein (Wuhan-Hu-1, MN908947) and T cell epitopes (TCE), the latter includes 77% of SARS-CoV-2 ORF3a, 79% of Nucleocapsid, and 18% of Membrane proteins. The GRT-R910 vector encodes the VEEV non-structural proteins as well as the 5′ and 3′ RNA sequences required for RNA replication, but encodes no structural proteins, and no infectious viral particles are formed. The VEEV sub-genomic 26S promoter is located 5′ of the inserted TCE cassette and this is followed by a second VEEV sub-genomic 26S promoter driving expression of the $S_{D614G}$ cassette. The $S_{D614G}$ protein was modified to replace the furin cleavage site at amino acids position 682-685 (RRAR) with a non-cleavable amino acid sequence (GSAS) and 2 proline amino acids substitutions at amino acid positions 986 and 987 (K986P, V987P)[53]. The samRNA vector (GRT-R910) was synthesized as previously described[20]. Briefly, $S_{D614G}$ and TCE sequences were PCR amplified and cloned into PacI/BstBI sites of a pUC02-VEE vector. Capped samRNA was synthesized in vitro using HiScribe T7 Quick High Yield RNA Synthesis Kit (New England Biolabs) and purified using a RNeasy Maxi Kit (Qiagen) according to the manufacturer's protocol. samRNA was subsequently encapsulated in an LNP using a self-assembly process in which an aqueous solution of samRNA is rapidly mixed with a lipid mixture in ethanol. RNA encapsulation efficiency was measured using Ribogreen RNA quantitation reagent (Thermo Fisher) and confirmed to be >95% in all batches analyzed. samRNA-LNP was formulated into a buffer containing 5 mM Tris (pH 8.0), 10% sucrose, 10% maltose.

## PBMC isolation and storage

Heparinized whole blood for cellular immunogenicity testing was collected prior to administration of the first GRT-R910 dose (Day 1) to assess baseline levels, and at days (D) 8, 29, 180, and 365 for participants receiving a single dose of GRT-R910 only. Blood collections for participants receiving an optional second dose of GRT-R910 (Table 1 and Fig. 1C) align with participants receiving single dose up to D29, and subsequent blood collections were at D113 (−3 days/+21 days) prior to administration of the second dose of GRT-R910, and at days 142 (4 weeks post dose), 293 (6 months post second dose), and 478 (12 months post second dose). PBMCs from whole blood were isolated at a local PBMC processing site according to standardized protocols. Briefly, cells were isolated using density gradient centrifugation on Ficoll® Paque Plus (GE Healthcare, Chicago, IL, USA), washed with D-PBS (Corning, Corning, NY, USA), counted, and cryopreserved in CryoStor CS10 (STEMCELL Technologies, Vancouver, BC, Canada) at $5 \times 10^6$ cells/ml. Cryopreserved cells were stored in liquid $N_2$ ($LN_2$), shipped in cryoports and transferred to storage in $LN_2$ upon arrival. Cryopreserved cells were thawed and washed twice in OpTmizer™ T Cell Expansion Basal Medium (Gibco, Gaithersburg, MD, USA) with Benzonase (EMD Millipore, Billerica, MA, USA) and once without Benzonase. Cell counts and viability were assessed using the Guava ViaCount reagents and module on the Guava EasyCyte HT cytometer (EMD Millipore). Cells were rested overnight prior to use in functional assays. D113 and D142 treatment days per protocol −3/+20 days.

Participants 0005, 0008, 0009, 0016, and 0023 were excluded at D293 and participant 0019 was excluded at D142 and D293 due to SARS-CoV-2 infections at days 167, 243, 248, 256, 276, and 123, respectively. Participants 0014 and 0024 excluded at D180 due to SARS-CoV-2 infection at days 107 and 123, respectively. Participant 0020 was excluded at D180 due to receiving mRNA BNT162B2 vaccine at day 114 and SARS-CoV-2 infection on day 237.

## Serum collection and storage

Whole blood for humoral immunogenicity testing was collected prior to administration of the first GRT-R910 dose (day 1) to assess baseline levels, and at days 29, 57, 180, and 365 for participants receiving a single dose of GRT-R910 only. Blood collections for participants receiving an optional second dose of GRT-R910 (Table 1) align with paticipants receiving single dose up to D57, and subsequent blood collections were at D113 (−3 days/+21 days) prior to administration of the second dose of GRT-R910, and at days 142 (4 weeks post dose), 293 (6 months post second dose), and 478 (12 months post second dose). Serum was collected, aliquoted, and cryopreserved and stored at −80 °C until shipment. Cryopreserved serum aliquots were shipped in cryoports and transferred to storage at −80 °C upon arrival. Serum sample aliquots were thawed prior to assessment of binding or neutralizing antibody assays as described below. Participants 0005, 0008, 0009, 0016, and 0023 are excluded at D293 and participant 0019 is excluded at D142 and D293 due to SARS-CoV-2 infections at days 167, 243, 248, 256, 276, and 123, respectively. Participants 0014 and 0024 are excluded at D180 due to SARS-CoV-2 infection at days 107 and 123, respectively. Participant 0020 is excluded at D180 due to receiving mRNA BNT162B2 vaccine at day 114 and SARS-CoV-2 infection on day 237.

## IgG ELISA and microneutralization assay (MNA) (VisMederi Srl)

Serum samples were sent to VisMederi Srl (Sienna, Italy) for independent analysis for anti-Spike (S) IgG levels via ELISA and neutralizing antibody titers via microneutralization assay (MNA) according to local standard operating procedures (SOPs). Briefly, SARS-CoV-2 prefusion $S_{WT}$ (Wuhan-Hu-1; MN908947) protein was coated onto 96-well microplates. Following incubation, the microplate is washed to remove unbound antigen and blocked to prevent non-specific binding. Standard, control, and sample dilutions are incubated in the coated microplate, in which anti-SARS-CoV-2 $S_{WT}$ IgG specific antibodies (primary antibodies) bind to the coated antigen. Following incubation, the microplate is washed to remove unbound primary antibodies. Primary antibodies are detected with the addition of the anti-human IgG antibody (secondary antibody) conjugated to peroxidase. After incubation, the microplate is washed to remove unbound secondary antibodies. The peroxidase substrate solution, tetramethylbenzidine (TMB), is added to the microplate and a colored product is developed which is proportional to the amount of SARS-CoV-2 Spike$_{WT}$ IgG antibodies present in the serum sample. 2 N $H_2SO_4$ is then added to stop the colorimetric reaction. The absorbance of each well is measured using a microplate spectrophotometer reader at a specific wavelength (450/620 nm). A standard on each tested plate is used to calculate the amount of antibodies against SARS-CoV-2 S in the sample according to the unit assigned by the standard (ELU/ml). The MNA is used to measure virus-specific neutralizing antibodies. Infectious virus is incubated with test substances, usually sera. Virus susceptible monolayers (Vero/E6 Cells) are exposed to the test substance/virus mixture. Patches of infected cells (microplaques) are detected via an antibody pair specific for the SARS-CoV-2 (Victoria/01/2020 strain; $S_{VIC}$ MT007544.1) RBD $S_{VIC}$ protein and visualized using TrueBlue™ substrate. Neutralizing titer of test samples is determined against an interna reference sera (Public Health England). $ND_{50}$ (neutralizing dilution at 50%) is reported.

## Mesoscale discovery (MSD) S and N IgG assay

Anti-S IgG levels were also assessed in-house through use of a V-Plex COVID-19 Serology kit from Mesoscale Discovery (MSD). Serum samples were tested according to manufacturer's instructions. Briefly, plates pre-coated with antigens of interest on spots in each well are blocked to prevent non-specific binding. Diluted samples, standards, and controls are added to each plate, and, after incubation, plates are washed to remove excess sample. Antigen-specific antibody contained within the sample is then detected using antibodies conjugated to MSD SULFO-TAG™. After incubation, excess detection antibody is removed, and MSD Read Buffer is added to the plate before reading on an MSD instrument (Meso Sector S 600). Emitted light from the SULFO-TAG is measured and sample concentration is determined using Methodical Mind acquisition software and MSD Discovery Workbench for data analysis. Concentration is reported as MSD arbitrary units (arb. units)/mL.

## Pseudovirus neutralization assay (PNA)

Pseudotyped virus particles were made using a genetically modified Vesicular Stomatitis Virus from which the glycoprotein G was removed (VSVΔG). The VSVΔG virus was transduced in HEK293T cells previously transfected with the Spike glycoprotein of the SARS-CoV-2 coronavirus (multiple strains: $S_{D614G}$, $S_{Beta}$, $S_{Delta}$, $S_{OmicronBA1}$ and $S_{OmicronBA5}$) for which the last 19 amino acids of the cytoplasmic tail were removed (ΔCT). The generated pseudovirus particles (VSVΔG – Spike ΔCT) for SARS-CoV-2 strains $S_{D614G}$, $S_{Beta}$, $S_{Delta}$, $S_{OmicronBA1}$ and $S_{OmicronBA5}$ expressed a luciferase reporter which can be quantified in relative luminescence units (RLU). Heat-inactivated serum samples were serially diluted (9-serial, 2-fold dilution) in a 96-well plate, in duplicate, and a pre-determined amount of pseudotyped virus (corresponding to 200,000 RLU/well) was applied to the plate and incubated with serum/plasma to allow binding of the neutralization antibodies to the pseudotyped virus. After the incubation of the serum/plasma-pseudotyped virus complex, the serum/plasma-pseudotyped virus complex was transferred to the plate containing Vero-E6 cells (ATCC). Test plates were incubated at 37 °C with 5% $CO_2$ overnight. Luciferase substrate was added to the plates which were then read using a plate reader detecting luminescence. The intensity of the light being emitted is inversely proportional to the amount of anti-SARS-CoV-2 neutralizing Spike antibodies bound to the VSVΔG – S ΔCT particles. Each microplate was read using a luminescence microplate reader with SoftMax Pro software v6.5.1 (SpectraMax). The dilution of serum required to achieve 50% infectious dose ($ID_{50}$) when compared to a non-neutralized pseudoparticle control was calculated for each sample dilution (average of duplicates) and the $ID_{50}$ is interpolated from a linear regression using the two dilutions flanking the 50% neutralization. Data analysis was performed using R Studio version 1.3.1093 and graphed using GraphPad Prism 9.0.1.

## Peptides

Custom-made, recombinant, lyophilized peptides (Supplementary Dataset 4) specific for each patient were produced by Genscript (Piscataway, NJ, USA) and reconstituted at 1 or 2 mg/ml/peptide in sterile 20% (v/v) $H_2O$ and 80% (v/v) DMSO (VWR International, Pittsburgh, PA, USA), aliquoted and stored at −80 °C. Control peptides to assess responses to infectious disease antigens from CMV, EBV, Influenza (CEF peptide pool) were purchased from JPT Peptide Technologies (Berlin, Germany).

## In vitro stimulation (IVS) cultures

Antigen-specific T cells from participant samples were expanded in the presence of peptide pools (Supplementary Dataset 4) and low-dose IL-2 as described previously[24]. Briefly, thawed PBMCS were rested overnight and stimulated in the presence of long overlapping (15mer) peptide pools (4 µg/ml/peptide) or control peptides (CEF) in

ImmunoCult™-XF T Cell Expansion Medium (IC media; STEMCELL Technologies) with 10 IU/ml rhIL-2 (R&D Systems Inc., Minneapolis, MN, USA) for 14 days in 48- or 24-well tissue culture plates. Cells were seeded at $1–2 \times 10^6$ cells/well and fed every 2–3 days by replacing 2/3 of the culture media with rhIL-2.

### IFNγ ELISpot assay

Detection of IFNγ-producing T cells was performed by ex vivo ELISpot assay[54]. Briefly, cells were harvested, counted and re-suspended in media at $4 \times 10^6$ cells/ml (ex vivo PBMCs) or $2 \times 10^6$ cells/ml (IVS-expanded cells) and cultured in the presence of 20% (v/v) $H_2O$ and 80% (v/v) DMSO (VWR International), Phytohemagglutinin-L (PHA-L; Sigma-Aldrich, Natick, MA, USA), CEF peptide pool, or peptide pools (Supplementary Dataset 4) in ELISpot Multiscreen plates (EMD Millipore) coated with anti-human IFNγ capture antibody (1:100 dilution, Mabtech, Cincinnati, OH, USA). Following 18-24 h incubation in a 5% $CO_2$, 37 °C, humidified incubator, supernatants were collected, cells were removed from the plate, and membrane-bound IFNγ was detected using anti-human IFNγ detection antibody (1:500, Mabtech), Vectastain Avidin peroxidase complex (Vector Labs, Burlingame, CA, USA) and AEC Substrate (BD Biosciences, San Jose, CA, USA). ELISpot plates were allowed to dry, stored protected from light, and read on AID plate reader (AID iSpot, Autoimmun Diagnostika GmbH, Straßberg, Germany) for standardized evaluation using the SoftMax Pro software v6.5.1 (SpectraMax). Data are presented as spot forming units (SFU) per million cells. Total ex vivo Spike response is calculated by the sum of responses to overlapping peptide (OLP) pools S 1-2, S 1-2, S 2-1, and S 2-2. Ex vivo ELISpot limit of detection (LOD) was determined at 30 SFU/$10^6$ PBMCs. T cell epitope (TCE) post-IVS ELISpot responses is calculated by the sum of responses to OLP pools covering N, M and ORF3a regions of the TCE cassette (Fig. 1A). Post-IVS ELISpot LOD was determined at 180 SFU/$10^6$ PBMCs. For both ex vivo and post-IVS ELISpot assay, if vehicle responses were above LOD, positive responses were determined at ≥2-fold increase compared to vehicle control.

### Intracellular cytokine staining (ICS)

IVS-expanded PBMCs were stimulated with either minimal (8-11mer) or long overlapping (15mer) peptides, 20% (v/v) $H_2O$ and 80% (v/v) DMSO (vehicle control; VWR) or PMA/Ionomycin cell stimulation cocktail (positive control; Affymetrix, Santa Clara, CA, USA) in the presence of anti-Human CD28/CD49d antibody (BD Biosciences, San Jose, CA, USA), BD GolgiStop (BD Biosciences), and Brefeldin A (BioLegend) over a period of 18 h. Following overnight incubation, cells were stained with live/dead Zombie-Red (BioLegend, 1:500 dilution) and surface marker antibodies (CD8-PerCP-Cy5.5 (1:200), CD3-BV605 (1:100), from BioLegend; CD4-APC-eF780 (1:100) from eBioscience) prior to fixation and permeabilization with FIX & PERM™ Cell Permeabilization Kit (ThermoFisher, Waltham, MA, USA). Following permeabilization, cells were stained for intracellular cytokines using anti-human IFNγ-APC (1:50), TNFα-BV785 (1:50), and IL-2-PE (1:25) antibodies (BioLegend) prior to data acquisition on a BD LSRFortessa™ flow cytometer (BD Biosciences) using FACSDiva™ Software version 9.2.

### Flow cytometry analyses

Samples acquired on flow cytometers were analyzed using FlowJo software (FlowJo, LLC, Ashland, OR, USA). Gating strategies for human samples (Supplementary Fig. S5) are as follows: Lymphocytes (SSC-A vs FSC-A), single cells (FSC-H vs FSC-A), viable cells (FSC-A vs LD-ZombieRed), CD3+ cells (FSC-A vs CD3-BV605), CD4+ and CD8+ (CD4-APCeF780 vs CD8-PerCP-Cy5.5), CD8+ or CD4+ cytokine+ cells (CD8-PerCP-Cy5.5 vs IFNγ-APC, TNFα-BV786, IL-2-PE; CD4-APCeF780 vs IFNγ-APC, TNFα-BV786, IL-2-PE). Polyfunctionality analyses were performed using Boolean gating (FlowJo Version 10.6.1.) and graphed using GraphPad Prism 9.0.1. Results are represented as % positive cell populations (frequency of parent). Data shown as background subtracted where indicated.

### DNA extractions

Isolated PBMCs (1–5 million) were thawed on ice prior to lysis and total nucleic acid extraction with the AllPrep DNA/RNA (Qiagen, Germany) coextraction kit following manufacturer's instructions. Genomic DNA yields were quantified with the Quant-iT PicoGreen dsDNA assay (ThermoFisher, USA).

### TCRb chain sequencing and analysis

Extracted genomic DNA from PBMCs was subjected to immunosequencing using the immunoSEQ Assay (Adaptive Biotechnologies, WA, USA). Primer sets designed to amplify V and J gene segments of the human TCRb gene in an initial round of PCR include 5′ universal adaptor sequences to serve as priming sites in a second "nested" PCR reaction where sample specific molecular barcodes and sequences required for next-generation sequencing (NGS) are incorporated. Synthetic TCRb molecules at known concentrations are included in the PCR to allow for quantification of a given sampled TCRb sequence[55]. The sequenced TCRb repertoires from the molecular assay can be ascribed to SARs-CoV-2 derived MHC complexed peptides and/or COVID-19 infection associated public TCRb sequences[56–58]. TCR breadth was assessed as the number of Covid-associated rearrangements out of the total number of rearrangements in a sample. TCR depth was assessed as the number of Covid-associated T cells (or templates) out of the total number of T cells (templates) in a sample.

### Control samples

Serum samples collected before the year 2020 and SARS-CoV-2 convalescent donor serum purchased from PFM (Precision for Medicine, Bethesda, MD, USA; Supplementary Table S1) were used to identify Nucleocapsid IgG antibody ranges for non-exposed individuals and SARS-CoV-2 convalescent individuals. PBMC samples processed from leukopaks in-house before the year 2020 or purchased from PFM (also processed pre-2020; Supplementary Dataset 6) were used to test TCE and Spike peptide pools in ex vivo ELISpot and post-IVS ELISpot for cross-reactive responses.

### Reporting summary

Further information on research design is available in the Nature Portfolio Reporting Summary linked to this article.

## Data availability

Pseudonymized individual participant clinical data that underlie the results reported in this article are, to the extent permitted by applicable data protection laws, including the UK GDPR and the UK Data Protection Act, available for transfer. Interested investigators can obtain and certify the data transfer agreement (DTA) and submit requests to the principal investigator K.J. Principal investigator K.J. will reply to requests within 2 months. Investigators and institutions who consent to the terms of the DTA form, including, but not limited to, the use of these data for the purpose of a specific project and only for research purposes, and to protect the confidentiality of the data and limit the possibility of identification of participants in any way whatsoever for the duration of the agreement, will be granted access. Gritstone will then facilitate the transfer of the requested pseudonymized data. This mechanism is expected to be via a Gritstone Secure File Transfer Service, but Gritstone reserves the right to change the specific transfer method at any time, provided appropriate levels of access authorization and control can be maintained. Source data are, to the extent permitted by applicable data protection laws, including the UK GDPR and the UK Data Protection Act, provided with this paper. Source data are provided with this paper.

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

## Acknowledgements

We would like to thank the study participants and their families, clinical staff and study coordinators, and the Coalition for Epidemic Preparedness Innovations (CEPI) for providing assay support via VisMederi Srl. Special thanks go to: Christopher Green, Adrian Palfreeman, Tolga Turgut, Valentina Bernasconi, Joshua Perez, Alexis Mantilla, Rita Zhou, Juan Betular, Laura Veckerelli, Ellen Smith, Katrina Chan, Jonathan Dayao, James Sun, Minh Duc Cao, Roman Yelensky, Justin Helbert, Orod Osanlou, Robert L. Veres, Elena Grillo, Uzma Ahmed, Ankur Dhanik, Raphael Rousseau, and members of Clinical Operations and Translational Medicine Operations teams who contributed to this study. Funding was provided by the study sponsor, Gritstone bio, Inc. Employees of Gritstone bio, Inc. received salaries and stock options. No other authors received direct funding for this work from the sponsor.

## Author contributions

D.O.K., K.J., C.D.P., and A.Us: Designed the study & immune monitoring approach. A.Ur and A.Us: contributed to clinical oversight, subject recruitment, enrollment, and treatment. C.D.S., L.G., K.J., A.R.R., J.K., J.H., and S.-J.H.: Contributed to vaccine design. L.D.K.T., M.A.K., C.D.P., A.R.R., M.G.H., C.N.N., M.M., S.K., H.V., J.R.J., T.Y., K.L., M.E., S.G., and M.J.D.: Contributed to immune data generation, assay development, data analyses, and methods. K.B., J.C.K., P.G., K.J., and C.D.P.: Contributed to safety data analyses. C.D.P., L.D.K.T., M.A.K., K.J., P.G., D.O.K., K.B., and A.Us: Contributed to manuscript writing.

## Competing interests

This study was sponsored by Gritstone bio, Inc. C.D.P., C.D.S., D.O.K., L.D.K.T., A.R.R., M.A.K., L.G., J.K., M.J.D., H.V., M.G.H., J.R.J., S.K., M.M., C.N.N., K.B., T.Y., K.L., M.E., S.-J.H., J.C.K., P.G., and K.J. are stockholders and either current or previous employees at Gritstone bio, Inc. and may be listed as co-inventors on various pending patent applications related to the vaccine platform presented in this study. D.O.K. is currently an employee and shareholder of Gilead sciences. P.G. holds shares in AstraZeneca and Takeda. The remaining authors declare no competing interests. Participants did not receive compensation for participating in this study.
