## [Peer Review File · Nature Communications]

REVIEWER COMMENTS

Reviewer #1 (Remarks to the Author):

The manuscript by Palmer et al details clinical assessment of GRT-R910 a self-amplifying mRNA vaccine encoding spike protein and NP, M and ORF-3 as a strategy to boost the breadth of the response. The authors demonstrate acceptable safety and reactogenicity when used as a booster vaccine in individuals over the age of 60, a priority population for annual COVID booster vaccines. However the manuscript is largely descriptive and lacks statistical analysis of immunology data. The main claim of increasing breadth of spike response is based on comparing late memory responses induced by AZD1222 to peak GRT-R910 responses, therefore not a valid comparison. Given the amount of preclinical and clinical papers documenting spike T cell and antibody responses in AZD122 primed and saRNA or mRNA boosted individual, this manuscript only marginally increases current knowledge.

- 1) There is a clear lack of statistical analysis of data, statements on increase in response are not supported by statistical analysis. This paper would significantly benefit from consultation from a statistician for more in depth analysis of the data to compare vaccine group/number of vaccines accounting for volunteers as a variable (eg repeated measures anova).
- 2) Statements about fold change in Ab levels due to vaccination are performed on a group basis, however volunteers will have different starting levels of antibodies. To analyse fold increase following vaccination, the authors may want to consider calculating fold increase compared to baseline for each individual and presentation of the data as a separate graph, this would give a clear demonstration of the increase post-vaccination and how much the response is maintained.
- 3) Authors claim an increase in breadth of the response following GRT-R910 vaccination, however it is not a valid analysis as they are comparing to pre-vaccination low level memory responses to high peak post-vaccination responses. The indication of increased breadth could simply be due to higher frequency of antigen specific cells above background at peak post-vaccination timepoint. Breadth analysis should also be performed at the 6 month timepoint, this would be a true comparison as to whether breadth has increased following vaccination. Only valid conclusion is that at the peak of the response broad spike specific response are detected, indicated by response to large number of spike peptide pools and TCR CDR3.
- 4) The main difference between cohorts is dose, however within the cohorts is a sub-group of those individuals that received 1 dose or 2 doses, with data split in figures to compare these sub-groups, but there is no statistical analysis to determine whether the second dose is beneficial. It would seem only 1 dose of vaccine is required. Authors should consider a formal analysis, particularly as data is presented to show these two groups.
- 5) The novel aspect of the paper is ability to induce response to other spike proteins where the T cell responses are much higher than spike. Because there are pre-existing responses (in some volunteers) and data isn't matched to individual volunteers, it is difficult to determine how of the T cell response is

induced or boosted by GRT-R910. The authors may want to consider calculating fold change from baseline and presentation as additional graph to demonstrate to readers what impact vaccination has on these responses (this may be able to identify boosted responses vs de-novo responses more clearly).

Minor comments

“Optimal second dose” – I would be cautious with the use of optimal, I understand the ideal vaccination regimen is 2 doses, however “did not receive an optimal second” could imply the volunteer had a mis-injection or not complete dose on the 2nd administration, as opposed to received just 1 dose of vaccine

It’s disappointing the authors have failed to acknowledge published preclinical and clinical studies assessing immunogenicity of saRNA or mRNA following priming with AZD122. These studies compared T cell (and breadth of spike response) and antibody responses and would be beneficial to compare whether the same pattern of responses are observed.

Figure 1 references cohort 3 and 4, for clarity would be good to statement about trial groups and why data from these volunteers has been not been included in this manuscript

Figure 2 to 5- I like the way different sub-groups are represented by different symbols colouring, the authors may want to consider adding a legend to the figure to indicate what the different colouring and symbols relate to. I struggled a bit to see the difference until I zoomed into figure, maybe consider using the same colour for outline as for fill.

Reviewer #2 (Remarks to the Author):

This is primary a statistical review, and the use of statistics, as noted by the authors, is only for descriptive purposes, so my comments are correspondingly minor. A few notes below.

* Were the current design to be extended into a trial intended to evaluate effects of second 10ug or 30ug doses, the protection from possible confounding offered by randomization would break down, since participants are allowed to opt in to second doses. In the current study, only descriptive statistics are provided as to IgG and nAb response to said doses, and no statements seem to be made about comparisons between arms; future work will likely make such comparisons, though.

* At the end of the supplementary section "Safety assessment", the authors mention the use of Pearson correlation, the Kruskal-Wallis test, and Dunn's multiple comparisons correction. It would be helpful to briefly note (simultaneously) exactly what these tests are used for, so that the interested reader is aware prior to when next they are mentioned.

* In lines 454-455, the authors note that T-cell data included some values "equals zero" (should be "equal to zero"), leading them to present these summaries based on arithmetic means rather than geometric means used for nAb and spike IgG data. It would seem more straightforward to instead modify the T-cell data by including an offset (e.g., adding 1 or another small number) and then using the geometric mean, making the results comparable in scale between T-cell, nAb, and spike IgG summaries.

REVIEWER COMMENTS

Reviewer #1 (Remarks to the Author):

The manuscript by Palmer et al details clinical assessment of GRT-R910 a self-amplifying mRNA vaccine encoding spike protein and NP, M and ORF-3 as a strategy to boost the breadth of the response. The authors demonstrate acceptable safety and reactogenicity when used as a booster vaccine in individuals over the age of 60, a priority population for annual COVID booster vaccines. However the manuscript is largely descriptive and lacks statistical analysis of immunology data. The main claim of increasing breadth of spike response is based on comparing late memory responses induced by AZD1222 to peak GRT-R910 responses, therefore not a valid comparison. Given the amount of preclinical and clinical papers documenting spike T cell and antibody responses in AZD122 primed and saRNA or mRNA boosted individual, this manuscript only marginally increases current knowledge.

- We thank the reviewer for their constructive comments and have provided detailed responses below. We acknowledge the limitation of this study due to the small sample size, which we have stated in the discussion. This phase 1 study was designed to be hypothesis generating rather than statistically powered for testing, which led to the use of descriptive rather than inferential statistics. We have, however, additionally analyzed the data using non-parametric statistics, which are more appropriate for small data sets. However, the results are meant to be supportive rather than conclusive.
- We would like to note that, while there is a plethora of human clinical data on mRNA vaccines in the literature, as far as we are aware, this is the first study assessing a self-amplifying mRNA (samRNA)-based SARS-CoV-2 vaccine as a boost following a primary series with an adenoviral vaccine in humans. The most notable differentiator between mRNA and samRNA vaccine platforms is the replication of samRNA within the transduced target cells, driving prolonged antigen expression, increased durability of nAb titers, and offering dose sparing potential in mice (Vogel, Lambert et al. 2018, de Alwis, Gan et al. 2021). We have previously shown that our samRNA-based vaccine induced potent nAb titers and provided protection from infection at a 10µg dose in non-human primates (NHPs), similar to levels described with 10-fold higher doses (100µg) of authorized SARS-CoV-2 mRNA vaccines in NHPs (Corbett, Flynn et al. 2020, Vogel, Kanevsky et al. 2021, Rappaport, Hong et al. 2022), further highlighting the dose-sparing potential of this novel vaccine platform.
- We have also updated our analyses of breadth comparing late memory post AZD1222 (5-7 months after vaccination) with late memory post GRT-R910 (6 months after vaccination) to enable a more accurate comparison.

1) There is a clear lack of statistical analysis of data, statements on increase in response are not supported by statistical analysis. This paper would significantly benefit from consultation from a statistician for more in depth analysis of the data to compare vaccine group/number of vaccines accounting for volunteers as a variable (eg repeated measures anova).

- We appreciate the reviewer's concerns and have added additional statistics where appropriate.
- Given the small sample size, mostly descriptive and non-parametric statistics were used for inference as supportive results.
- Wilcoxon signed rank test statistic was used to test whether changes in immune responses from baseline were statistically significant or not. This test is appropriate when data are paired (data from two different time points from the same subject).
- Mann-Whitney test was used to test whether the 30µg dose of samRNA increased IgG and nAb levels further compared to the 10µg dose. This test is appropriate for non-paired data.
- Categorical data were tested by Chi-Square test.
- Descriptive statistics for fold change from baseline have been added to supplementary figures and tables, as well as the relevant text describing the antibody data.
- Fold change comparisons of T cell data at 4-week and 6-month time points post GRT-R910 administration analogous to Liu *et al.* (performed for 4 weeks versus 3-month timepoints (Liu, Munro et al. 2022)) have been included in the supplementary materials.
- Additional figures outlining fold change of antibody responses per individual have been added to the supplemental material as requested (see reviewer's comments 2 & 5 below).

2) Statements about fold change in Ab levels due to vaccination are performed on a group basis, however volunteers will have different starting levels of antibodies. To analyse fold increase following vaccination, the authors may want to consider calculating fold increase compared to baseline for each individual and presentation of the data as a separate graph, this would give a clear demonstration of the increase post-vaccination and how much the response is maintained.

- We would like to thank the reviewer for this insightful comment, the spider plot for fold change over baseline of nAb at each time point did indeed reveal interesting observations both with regards to the boosting effect of the second dose at ~4 months and maintenance of the response through 6 months post one or two doses of GRT-R910. We have added these analyses to new Supplemental Figures S3 and S4 and have also provided summary statistics of the fold change by nAb against ancestral Spike and VOCs beta, delta, omicron BA1 and BA5 in new Supplementary Tables S2A and S2B. The relevant sections of the results and discussion, as well as figure legends, have been updated accordingly.

3) Authors claim an increase in breadth of the response following GRT-R910 vaccination, however it is not a valid analysis as they are comparing to pre-vaccination low level memory responses to high peak post-vaccination responses. The indication of increased breadth could simply be due to higher frequency of antigen specific cells above background at peak post-vaccination timepoint. Breadth analysis should also be performed at the 6 month timepoint, this would be a true comparison as to whether breadth has increased following vaccination. Only valid conclusion is that at the peak of the response broad spike specific response are detected, indicated by response to large number of spike peptide pools and TCR CDR3.

- The reviewer makes a valid point, and 6-month breadth data for Spike and TCE T cell responses have been added to Figures 4 and 5, respectively, and the wording has been adjusted accordingly.
- Of note, the Day 1 (baseline) Spike pool responses measured are 19-31 weeks (5-7 months) post the last AZD1222 vaccination, showing breadth of post AZD1222 memory response as the reviewer points out. However, in comparison, our data at 6 months post GRT-R910 administration suggest either prolonged boosting of previously primed responses, or indeed increased breadth, as 66% of subjects have measurable T cell responses to 3 or more Spike pools 6 months following GRT-R910 administration compared to 25% 5-7 months following AZD1222 administration.
- We would also like to point out that we discovered an error in the T cell data from two subjects, where the values included in graphs had not been converted to SFU/10⁶ cells, and hence were off by a factor of 5 (raw data are SFU/2x10⁵ cells). T cell figures and tables have been updated with the correct values.

4) The main difference between cohorts is dose, however within the cohorts is a sub-group of those individuals that received 1 dose or 2 doses, with data split in figures to compare these sub-groups, but there is no statistical analysis to determine whether the second dose is beneficial. It would seem only 1 dose of vaccine is required. Authors should consider a formal analysis, particularly as data is presented to show these two groups.

- We have addressed this valid point by comparing fold change over baseline over time for each individual comparing participants receiving one or two doses in new Supplementary Figures S3 (IgG) and S4 (nAb), and have added the summary statistics in new Supplementary Tables S2A and S2B.

5) The novel aspect of the paper is ability to induce response to other spike proteins where the T cell responses are much higher than spike. Because there are pre-existing responses (in some volunteers) and data isn't matched to individual volunteers, it is difficult to determine how of the T cell response is induced or boosted by GRT-R910. The authors may want to consider calculating fold change from baseline and presentation as additional graph to demonstrate to readers what impact vaccination has on these responses (this may be able to identify boosted responses vs de-novo responses more clearly).

- We appreciate the reviewer’s comments, and have clarified the data shown in Figure 5B to clearly separate participants with pre-existing T cell responses to Nuc, Mem and ORF3a (positive at baseline and positive after GRT-R910 administration) versus those without pre-existing responses, where administration of GRT-R910 has primed *de novo* responses (negative at baseline and positive after GRT-R910 administration) analogous to previous prime versus expansion analyses performed for neoantigen-specific T cells (Palmer, Rappaport et al. 2022).
- Given the presence of higher background activation in this assay (post-IVS ELISpot), fold-change analyses do not seem appropriate for these data, which are presented as background subtracted and indicated as positive or negative based on raw data stimulation values above or below LOD (in cases where DMSO control was <LOD) and whether values were < or ≥2-fold increased compared vehicle control (where vehicle responses were above LOD). So, in instances where a baseline timepoint has high background (DMSO > LOD), but the corresponding post GRT-R910 administration timepoints do not have high background (DMSO < LOD), fold change calculations would not accurately reflect changes in T cell responses.
- As noted above for the Spike data, the error in SFU values for two subjects has also been corrected for TCE values, and T cell figures and tables have been updated accordingly.

Minor comments

“Optimal second dose” – I would be cautious with the use of optimal, I understand the ideal vaccination regimen is 2 doses, however “did not receive an optimal second” could imply the volunteer had a mis-injection or not complete dose on the 2nd administration, as opposed to received just 1 dose of vaccine

- We are not sure which text the reviewer is referring to here, as we do not refer to an optimal second dose, but an *optional* second dose for cohorts 1 and 2 throughout the text.

It’s disappointing the authors have failed to acknowledge published preclinical and clinical studies assessing immunogenicity of saRNA or mRNA following priming with AZD122. These studies compared T cell (and breadth of spike response) and antibody responses and would be beneficial to compare whether the same pattern of responses are observed.

- We apologize for this oversight and have expanded our discussion on a comparable mRNA boost study following primary vaccination series with adenoviral vaccines (Munro, Janani et al. 2021, Liu, Munro et al. 2022) and have added clinical and preclinical self-amplifying RNA references (de Alwis, Gan et al. 2021, Low, de Alwis et al. 2022, Rice, Verma et al. 2022) to the discussion as well.

Figure 1 references cohort 3 and 4, for clarity would be good to statement about trial groups and why data from these volunteers has been not been included in this manuscript

- We have clarified this on page 4.

Figure 2 to 5- I like the way different sub-groups are represented by different symbols colouring, the authors may want to consider adding a legend to the figure to indicate what the different colouring and symbols relate to. I struggled a bit to see the difference until I zoomed into figure, maybe consider using the same colour for outline as for fill.

- We thank the reviewer for this input and have updated all figures, hopefully providing more clarity on the sub-groups.

Reviewer #2 (Remarks to the Author):

This is primarily a statistical review, and the use of statistics, as noted by the authors, is only for descriptive purposes, so my comments are correspondingly minor. A few notes below.

- We would like to thank the reviewer for their insightful feedback and have addressed their comments below.

* Were the current design to be extended into a trial intended to evaluate effects of second 10ug or 30ug doses, the protection from possible confounding offered by randomization would break down, since participants are allowed to opt in to second doses. In the current study, only descriptive statistics are provided as to IgG and nAb response to said doses, and no statements seem to be made about comparisons between arms; future work will likely make such comparisons, though.

- The reviewer makes a valid point regarding comparisons of participants receiving one versus two doses of GRT-R910. We have addressed this by comparing fold change from baseline over time for each individual comparing participants receiving one or two doses in new Supplemental Figures S3 and S4, and have also provided summary statistics in new Supplementary Tables S2A and S2B. The relevant sections of the results and discussion, as well as figure legends, have been updated accordingly.

* At the end of the supplementary section "Safety assessment", the authors mention the use of Pearson correlation, the Kruskal-Wallis test, and Dunn's multiple comparisons correction. It would be helpful to briefly note (simultaneously) exactly what these tests are used for, so that the interested reader is aware prior to when next they are mentioned.

- We appreciate the reviewer pointing this out, since the statistical analyses listed were relevant to the immune data (not safety data) and have been moved to the subsequent "Statistical Methods" section instead. In addition, we have added which data figures the specific tests apply to and have also updated the corresponding figure legends accordingly.

* In lines 454-455, the authors note that T-cell data included some values "equals zero" (should

be "equal to zero"), leading them to present these summaries based on arithmetic means rather than geometric means used for nAb and spike IgG data. It would seem more straightforward to instead modify the T-cell data by including an offset (e.g., adding 1 or another small number) and then using the geometric mean, making the results comparable in scale between T-cell, nAb, and spike IgG summaries.

- We thank the reviewer for this suggestion and have re-graphed the T cell data with values equal to zero set to $\frac{1}{2}$ the LOD for the relevant assay (*ex vivo* ELISpot LOD is 30 SFU/ 10^6 cells and post-IVS ELISpot LOD = 180 SFU/ 10^6 cells). Figures 4, 5, S6 and Supplementary Table S4 have been updated with Geometric means, and methods, figure legends, and relevant text sections have been updated accordingly.
- We would also like to point out that we discovered an error in the T cell data from two subjects, where the values included in graphs had not been converted to SFU/ 10^6 cells, and hence were off by a factor of 5 (raw data are SFU/ 2×10^5 cells). T cell figures and tables have been updated with the correct values.

References

Corbett, K. S., et al. (2020). "Evaluation of the mRNA-1273 Vaccine against SARS-CoV-2 in Nonhuman Primates." *N Engl J Med* **383**(16): 1544-1555.

de Alwis, R., et al. (2021). "A single dose of self-transcribing and replicating RNA-based SARS-CoV-2 vaccine produces protective adaptive immunity in mice." *Mol Ther* **29**(6): 1970-1983.

Liu, X., et al. (2022). "Persistence of immunogenicity after seven COVID-19 vaccines given as third dose boosters following two doses of ChAdOx1 nCov-19 or BNT162b2 in the UK: Three month analyses of the COV-BOOST trial." *J Infect* **84**(6): 795-813.

Low, J. G., et al. (2022). "A phase I/II randomized, double-blinded, placebo-controlled trial of a self-amplifying Covid-19 mRNA vaccine." *NPJ Vaccines* **7**(1): 161.

Munro, A. P. S., et al. (2021). "Safety and immunogenicity of seven COVID-19 vaccines as a third dose (booster) following two doses of ChAdOx1 nCov-19 or BNT162b2 in the UK (COV-BOOST): a blinded, multicentre, randomised, controlled, phase 2 trial." *Lancet* **398**(10318): 2258-2276.

Palmer, C. D., et al. (2022). "Individualized, heterologous chimpanzee adenovirus and self-amplifying mRNA neoantigen vaccine for advanced metastatic solid tumors: phase 1 trial interim results." *Nat Med*.

Rappaport, A. R., et al. (2022). "Low-dose self-amplifying mRNA COVID-19 vaccine drives strong protective immunity in non-human primates against SARS-CoV-2 infection." *Nat Commun* **13**(1): 3289.

Rice, A., et al. (2022). "Heterologous saRNA Prime, DNA Dual-Antigen Boost SARS-CoV-2 Vaccination Elicits Robust Cellular Immunogenicity and Cross-Variant Neutralizing Antibodies." Front Immunol **13**: 910136.

Vogel, A. B., et al. (2021). "BNT162b vaccines protect rhesus macaques from SARS-CoV-2." Nature **592**(7853): 283-289.

Vogel, A. B., et al. (2018). "Self-Amplifying RNA Vaccines Give Equivalent Protection against Influenza to mRNA Vaccines but at Much Lower Doses." Mol Ther **26**(2): 446-455.

REVIEWERS' COMMENTS

Reviewer #1 (Remarks to the Author):

Thank you all the work to revise the manuscript and address my comments, I am satisfied with the revised manuscript.

Reviewer #2 (Remarks to the Author):

The authors are to be commended for a timely and thorough revision of an important contribution. All of my concerns have now been addressed. comparable in scale between T-cell, nAb, and spike IgG summaries.